# Investigation of Accuracy of TOA and SNR of Radio Pulsar Signals for Vehicles Navigation

**DOI:** 10.3390/s23157010

**Published:** 2023-08-07

**Authors:** Hristo Kabakchiev, Vera Behar, Dorina Kabakchieva, Valentin Kisimov, Kamelia Stefanova

**Affiliations:** 1Faculty of Mathematics and Informatics, Sofia University, 1164 Sofia, Bulgaria; 2IICT, 1113 Sofia, Bulgaria; vera.behar@yahoo.com; 3Department of Information Technologies and Communications, University of National and World Economy, 1700 Sofia, Bulgaria; dkabakchieva@unwe.bg (D.K.); vkisimov@unwe.bg (V.K.); kstefanova@unwe.bg (K.S.)

**Keywords:** pulsar, radar, GPS, signal processing, navigation

## Abstract

It is known that X-ray and gamma-ray pulsars can only be observed by spacecraft because signals from these pulsars are impossible to be detected on the Earth’s surface due to their strong absorption by the Earth’s atmosphere. The article is devoted to the theoretical aspects regarding the development of an autonomous radio navigation system for transport with a small receiving antenna, using radio signals from pulsars, similar to navigation systems for space navigation. Like GNSS systems (X-ray and radio), they use signals from four suitable pulsars to position the object. These radio pulsars (out of 50) are not uniformly distributed but are grouped in certain directions (at least 6 clusters can be determined). When using small antennas (with an area of up to tens of square meters) for pulsar navigation, the energy of the pulsar signals received within a few minutes is extremely insufficient to obtain the required level of SNR at the output of the receiver to form TOA estimation, ensuring positioning accuracy up to tens of kilometers. This is one of the scientific tasks that is solved in the paper by studying the relationship between the SNR of the receiver output, which depends on the size of the antenna, the type of signal processing, and the magnitude of the TOA accuracy estimate. The second scientific task that is solved in the paper is the adaptation of all the possible approaches and algorithms suggested in the statistical theory of radars in the suggested signal algorithm for antenna processing and to evaluate the parameters of the TOA and DS pulsar signals, in order to increase the SNR ratio at the receiver output, while preserving the dimensions of the antenna. In this paper, the functional structure of signal processing in a pulsar transport navigation system is proposed, and the choice of the observed second and millisecond pulsars for obtaining a more accurate TOA estimate is discussed. The proposed estimates of positioning accuracy (TOA only, no phase) in an autonomous pulsar vehicle navigation system would only be suitable for the navigation of large vehicles (sea, air, or land) that do not require accurate navigation at sea, air, or desert. Large-sized antennas with an area of tens of square meters to hundreds of square meters can be installed in such vehicles.

## 1. Introduction

Since the middle of the last century, passive space radio navigation systems have been used for aircraft and submarine navigation by receiving the natural radio signals from the Sun and Moon emitted in space. This principle of passive radio navigation can also be proposed for space, air, marine, and ground vehicle navigation by working with other cosmic emitters such as pulsars.

The characteristic of pulsars is that they emit a series of pulses with very high stability of the pulse repetition period similar to or higher than that of the atomic clocks used in space satellite radio navigation systems—GNSS [1,2,3,4]. Although the individual pulses emitted by pulsars vary in terms of emitted power—amplitude and shape, the shape of the average pulse (profile) is stable and remains characteristic of each pulsar. 

For the past ten years, it is precisely these useful properties of pulsars that have been used to navigate spacecraft moving in space. This technology works always with stars present in the sky (day and night) and does not depend on human intervention. While in space this is the only alternative for navigation however, on Earth radio navigation with space satellite coverage—GNSS technologies—is mainly used for navigation [3,5,6,7,8,9,10,11,12]. 

In the USA, Russia, and China, pulsar navigation is mainly used to control space objects. In the event of a failure in the GPS GNSS, the X-ray pulsar navigation system becomes indispensable because it uses the sextant principle as suggested by the project of the US National Space Agency named Neutron-star Interior Composition Explorer/Station Explorer for X-ray Timing and Navigation Technology (NICER/SEXTANT). This project demonstrates an XNAV pulsar navigation system that is currently used on the spacecraft Artemis [13].

It is known that X-ray and gamma-ray pulsars can only be observed by space crafts because signals from these pulsars are impossible to be detected on the Earth’s surface due to their strong absorption by the Earth’s atmosphere. With the blocking of GPS GNSS, all transport ground and air means of transport (ships, planes) will remain without cosmic navigation and ground support. The military XNAV pulsar navigation system will not be available for civilian navigation. For these two reasons, the development of a terrestrial radio pulsar navigation system for transportation is a very current scientific and applied task that awaits its solution.

The national project of the University of National and World Economy (UNWE) BG05M2OP001-1.002-0002-C02 builds infrastructure and architectures for processing large volumes of data, as well as information communication technology and architecture for connectivity with IoT (Internet of Things) devices [14]. The project also develops principles, methods, and programs for working with sensors from cars and transport trucks. 

With this aim, the article is devoted to the theoretical aspects regarding the development of an autonomous transport radio navigation system with a small receiving antenna, using radio signals from pulsars, similar to navigation systems for space navigation. The transport navigation concept requires significantly reducing the observation time (within a few minutes), providing the transport positioning accuracy that is close to potential using B0329+54-type pulsars. The paper shows that in theoretical papers on space and aviation navigation published years ago, systems with small receiving antennas (10 sq.m) using pulsar signals have rather coarse positioning accuracy. Like GNSS systems, they use signals from 4 suitable pulsars (out of 50) to position the object. These pulsars are not uniformly distributed but are grouped in certain directions (at least six clusters can be determined). Such a pulsar is the pulsar B0329+54 researched in the article. In our research, we use the well-known rougher temporal (not phased) approach to transport navigation that was successfully used in early versions of GNSS, because, unlike artificial signals for positioning in GNSS, pulsar signals are not modulated and it is impossible to identify the number of phase cycles (pulsar periods) between two signal positions.

The transport navigation concept requires significantly reducing the observation time (within a few minutes), providing the transport positioning accuracy that is close to potential using B0329+54-type pulsars, with no large antenna. This is one of the scientific tasks that is solved in the paper by studying the relationship between the SNR of the receiver output, which depends on the size of the antenna, the type of signal processing, and the magnitude of the TOA accuracy estimate. The second scientific task that is solved in the paper is the adaptation of all the possible approaches and algorithms suggested in the statistical theory of radars in the suggested signal algorithm for antenna processing and to evaluate the parameters of the TOA and DS pulsar signals, in order to increase the SNR ratio at the receiver output, while preserving the dimensions of the antenna.

This challenge led us in the present article to think about what known statistical methods and signal processing approaches can be used, to maximize as much as possible the receiving pulsar energy at the input of the small antenna device, and to speed up and improve the quality of pulsar timing, for a few minutes. Since we work in the field of radar signal processing, which is designed to work in real time, we propose to use its concepts, approaches, and algorithms. 

**In Section 3**, theoretical estimates of positioning accuracy of space objects previously obtained by Sala’s team are as follows: 0.05 ∙ 10^6^ m—for 10 min of observation; 0.3 ∙ 10^5^ m. (50–30 km)—for 100 min of observation; 0.1 ∙ 10^5^ m—for 400 min of observation. Under all other ideal operating conditions of the space navigation system, these values of positioning accuracy are grossly inappropriate for real-time operation. The article proposes real (experimental) rough position estimates (in time only) of the TOA obtained using signal records from pulsar B0329+54. They are in the range from 24 km to 1 km with an observation time of 2–3 min. 

We first checked in the present paper (in Section 4 and Section 5) the quality of the signal algorithms proposed and developed by us for assessing the accuracy of TOA estimation presented in previous publications. The verification consisted of evaluating their proximity to the potential Cramer–Rao Lower Bound (CRLB) accuracy estimates, at the same SNR ratios as for the recorded pulsar signals. 

**In Section 4**, a potential TOA estimate was proposed to be calculated as a CRLB estimate, the calculation of which also requires an estimate of the SNR level and the parameters of the profile of the pulsar B0329+54, taken from the European pulsar database. 

From the results obtained **in Section 5**, it follows that the rough estimates of TOA obtained by us, at high levels of SNR are close to the potential CRLB estimates. They approach the CRLB estimates, which for the pulsar B0329+54 are from 1 to 10 nm, with large SNR (20–30 dB). The article states that it is necessary to look for opportunities to significantly reduce the size of the antenna system compared to those used in the experiment (500 m^2^ and 7000 m^2^) while ensuring a high level of SNR. 

**In Section 6**, the article is devoted to the theoretical aspects regarding the development of an autonomous transport radio navigation system with a small receiving antenna, using radio signals from pulsars, similar to navigation systems for space navigation. We only consider the issues of pulsar signal processing in the proposed functional structure for signal processing in a pulsar transport navigation system. Like GNSS systems, it uses signals from 4 suitable pulsars (out of 50) to position the object. More powerful second pulsars can be used, which can be observed with smaller antennas. The TOA measurement error for these pulsars is of the order of tens–hundreds of microseconds. The millisecond pulsars can be used with the measurement error of TOA less than 10 microseconds.

**In Section 7**, the paper suggests all the different approaches proposed in statistical radar theory to increase the receiver output SNR ratio while saving antenna dimensions. These approaches include space–time processing of signals in an antenna array, multi-beam reception of the various broadband pulsar signals for navigation (not less than four), and simultaneous reception of signals from one pulsar at different observation frequencies. Following this approach, we propose in this paper the simultaneous use of several antenna arrays, matched filtering with the shape of the pulsar signal in each channel of the linear antenna array, and multi-frequency simultaneous reception of the pulsar signals in these antenna arrays from the same pulsar. All this signal processing jointly increases the maximum possible SNR ratio at the output of the antenna system. 

In order to satisfy the requirements for real-time operation in the estimation of the parameters of pulsar signals (TOA and Doppler shift), we propose to use a multi-channel (parallel) approach. It allows at the same time to perform the estimation of the Doppler velocity of the vehicle in different frequency channels of the measurement frequency interval. In each of these frequency channels, TOA estimation is performed simultaneously in time channels of the measured time interval. 

The proposed estimates of positioning accuracy (TOA only, no phase) in an autonomous pulsar vehicle navigation system would only be suitable for the navigation of large vehicles (sea, air, or land) that do not require accurate navigation at sea, air, or desert. Large-sized antennas with an area of tens of square meters to hundreds of square meters can be installed in such vehicles.

## 2. Navigation with Pulsar Radio Signals

Pulsars are rapidly rotating neutron stars (Figure 1) [1,2,3,4]. Recently, there has been a particular interest in them due to their structure, their enormous density, strong magnetic and gravitational fields. Their matter is analogical to a huge atomic nucleus. A pulsar can be described as a huge, magnetized body rotating around its own axis, which does not coincide with the axis of the magnetic field. Since the magnetic axis of the pulsar does not coincide with the axis of its rotation, the stream of emitted radio waves spreads in space like a beam from a flashing beacon (a stream of radio pulses). What is special about this emission of pulsars is the high stability of the repetition period of the emitted pulses, comparable to that of the atomic clocks used in GPS navigation. 

This suggests that pulsar signals can be used in a system to navigate space and ground vehicles [1,2,3,4,5,6,7,8,9,10,11,12] similar to GNSS. Unfortunately, not all pulsars can be used for navigation, only those with a single-component mean pulse (profile). Most preferred for navigation are those pulsars whose single-component mean pulse (profile) is as narrow as possible at its peak. In this way, the highest accuracy of TOA measurement is ensured, and therefore so is the distance to the moving object and its coordinates [1,2]. 

The detailed analysis of most known pulsars, aiming to determine the most suitable of them for space navigation, was carried out in [2]. The analysis of pulsars is made according to several criteria concerning the energy of the emitted signals, the frequency of following pulses, the shape of the average pulse, etc. As a result of analysis, 50 pulsars have been identified, the signals of which are most suitable for space navigation. 

This number of pulsars is comparable to the total number of satellites used in modern GNSS (GPS, GLONAS, BeiDOU) [5]. The distribution of 50 pulsars in space elliptical and galactic coordinates, whose signals are most suitable for navigation is schematically illustrated in Figure 1 and Figure 2 [2]. As can be seen from Figure 1 and Figure 2, the distribution of pulsars covers almost the entire space, and, therefore, pulsars such as GNSS satellites can be used for space navigation [1,2]. These pulsars are not uniformly distributed but are grouped in certain directions (at least six clusters can be determined). 

Most X-ray pulsars emit very weak signals. The Crab pulsar (PSR B0531+21) is the most powerful pulsar in the X-ray range with a radiation flux density ~9.9 · 10^−9^ erg cm^−2^s^−1^ with photon energy 2–10 keV. Other pulsars in this range are much weaker emitters. It is known that X-ray and gamma-ray pulsars can only be observed by spacecraft because signals from these pulsars are impossible to be detected on the Earth’s surface due to their strong absorption by the Earth’s atmosphere.

On Earth’s surface, only radio pulsars can be used for navigation. In this case, more powerful second pulsars can be used, which can be observed with smaller antennas. However, these pulsars have a pulse emission period of the order of 1 s (second pulsars). The TOA measurement error for these pulsars is of the order of tens–hundreds of microseconds. The other group of pulsars is the so-called millisecond pulsars, the pulse repetition period of which is on the order of several milliseconds. However, these pulsars have less radiation power and therefore require antennas with a larger area for observation. However, the measurement error of TOA is less than 10 microseconds.

The schematic image of e/m emission from a pulsar is shown in Figure 3 [15], and Figure 4 illustrates a sequence of pulses emitted by pulsar B0329+54 [1,15]. The pulsar B0329+54, which is a neutron star, emits the pulse sequence with the e.m. energy spectral density S = 203 mJy at a frequency of 1400 MHz. The profile of the emitted pulses has the following parameters: (i)—repetition period T = 0.71452 s, (ii)—pulse width at 10% level T10 = 31.4 ms; (iii)—pulse width at 50% level T50 = 6.6 ms. These profile parameters presented in the EPN pulsar database are used in the CRLB estimates of the TOA. As can be seen from Figure 4, the individual pulses of radio pulsars do not resemble each other. After averaging a large number of individual emitted pulses (about 1000), the mean pulse profile is obtained. For different pulsars, the shape of the mean pulse profile is different and its mean width varies from 0.01 to 0.1 period [1,2]. 

A single pulse may contain several sub-pulses, which are usually symmetrical in shape and occur at random places inside successive pulses. In a number of pulsars, during the emission of pulses in each period, the shift of the sub-pulses from one end of the average pulse profile to its opposite end is observed. The shape of the average pulse can be one-component, two-component, or multi-component, depending on the distribution of sub-pulses.

For example, the average pulse profile of the pulsar B0329+54, which contains the main pulse and two sub-pulses, is shown in Figure 5 [1,2].

The following text is not related to navigation. However, it expands the known facts about pulsar B0329+54, and discovers new facts that may be useful to the wider scientific community in the field of signal processing, expanding the knowledge of the properties of pulsar signals. The obvious connection presented in Figure 5 and Figure 6 between the pulses of the human heart and the researched by us pulsar B0329+54, and a microscopic photo in Figure 7 and Figure 8 of a water drop irradiated with sound from pulsar B0329+54s the result of our research on the properties of pulsar signals in the last 5 years. 

Various Internet sites and publications liken pulsars to “radio beacons” and present them as the “hearts” of the universe. This comparison became popular after NASA’s mission with Voyager 1 and 2, and especially after the transformation of signals from pulsars (and other space objects) into sound signals, carried out by NASA and Roscosmos after 2020 [16]. The pulses emitted by pulsars resemble the pulses of the human heart. For comparison, in Figure 6. shows the ECG of a healthy human heart and its average impulse. A comparison of Figure 5 and Figure 6 shows the similarity of the two average pulses, those of the pulsar and the human heart [17].

In Bulgaria, in January 2022, the researcher Ivan Todorov [18] obtained in laboratory conditions digital records from a microscope of the structure of water drops irradiated by pulsar signals in the sound range. Using a methodology similar to that of Prof. Bernd-Helmut Kroeplin [19], Ivan Todorov has obtained different images of the structure of the water drop subjected to the impact of sound signals from different pulsars and kindly provided them to us for the purposes of this article. 

For example, two images in Figure 7 and Figure 8, obtained by him, show the microscopic structure of a water drop (or part of it) after irradiation of it by a signal from the pulsar B0329+54. Images were obtained at various microscope magnifications from 40 to over 1000 times. In Figure 7, the edge of the drop is shown at a low magnification of the microscope (about 40). The image of one of the characteristic elements of the structure of the edge of a water drop (Figure 7) is presented in Figure 8. This image was obtained on a specialized microscope at a very high magnification (over 5000–6000 times).

In 2014, Paul A. La Violette hypothesized that radio pulsars are beacons artificially created by extraterrestrial intelligence (ETI). They broadcast signals to various galaxies, including our solar system, with their main purpose being for interstellar navigation [20]. 

Our previous paper [21,22,23,24,25,26,27] used digital records of the signals obtained in the radio telescopes at the Dwingeloo and Westerbork observatories, which observe the pulsar B0329+54, for experimentally evaluating the accuracy of the TOA (time delay of the accumulated pulsar signal relative to the pulsar reference signal, i.e., template). 

## 3. Overview of Potential Navigational Accuracy by Using Pulsar Radio Signals 

During the Middle Ages, a man used the stars to navigate his travels by ship and later by airplane. In the 20th century, when there was no GNSS technology yet, the military used passive radio navigation systems operating on cosmic signals from the Sun and Moon (stars and planets). These systems were used to navigate nuclear-powered submarines, ships, and aircraft.

Commercial GPS jammers are now readily available on the Internet. A chain reaction of space debris, known as an ablation cascade, or a strong Earth-centered solar storm can interfere with GPS capability. Another serious drawback is the lack of accuracy in the subpolar regions of the Earth, caused by the low inclination of the orbits of GPS satellites (about 55 degrees). Without the use of GPS, the military ships at sea must find alternative ways to obtain an accurate fix on their position, and the military has just resumed training officers in the lost art of celestial navigation [28].

The U.S. Navy and Department of Defense take cyber threats to technology infrastructure seriously [28] and that is why they ordered companies Strategic and Spectrum Missions Advanced Resilient Trusted Systems (S2MARTS) to develop a navigation system that will push GPS to the second plan. In the event of a failure in the GPS GNSS, the X-ray pulsar navigation system becomes indispensable because it uses the sextant principle as suggested by the project of the US National Space Agency named Neutron-star Interior Composition Explorer/Station Explorer for X-ray Timing and Navigation Technology (NICER/SEXTANT). This project demonstrates an XNAV pulsar navigation system that is currently used on the spacecraft Artemis [13].

Currently, the Draper Laboratory scientists, in 2022, have patented a star-only space navigation system called a sliced-lens star tracker that can achieve 50-m accuracy in a non-GNSS-assisted environment [29]. Another possible modern application of pulsar signals is for the monitoring of airborne and space objects by using the forward scatter effect of a target irradiated with a pulsar signal when crossing a bistatic passive radio system [30].

The concept of using radio or (X-ray) pulsars for navigation is based on the measurement of pulse arrival times that are compared with the predicted time of arrival at a given location in a given era [31,32]. In this case, it is necessary to reduce the observed time of arrival of the pulsar pulse at the barycenter of the solar system and to correct the pulse arrival time. The parameters of this correction are the pulsar ephemeris together with the position and velocity of the observer relative to the barycenter of the solar system.

In [31], the concept of using radio pulsars for space navigation is proposed. Three options were considered for the receiving antenna for observing pulsars on board the spacecraft:—a multi-element phased antenna array (PAR), ~80 m^2^, spherical mirror with a diameter of 18−20 m; mirror parabolic antenna with a diameter of 9–10 m. The errors of high-precision measurement of the spacecraft coordinate in a transfer orbit to Mars using four pulsars PSR J1537 + 1155, J0437 − 4715, J1939 + 2134, J2145 − 0750 are measured. It is shown that in the most favorable case using high-precision measurement, the accuracy of measuring the spacecraft coordinates is 3 km up to 15 km. A rough determination of the position of the spacecraft in space is performed with an accuracy of tens and hundreds of kilometers [31].

It is known that X-ray and gamma-ray pulsars can only be observed by space crafts because signals from these pulsars are impossible to be detected on the Earth’s surface due to their strong absorption by the Earth’s atmosphere. 

In the event of a magnetic storm, radio jamming or other blocking of GPS GNSS, other GNSS radio navigation systems, as well as blocking of ground navigation systems, all transport ground and air means of transport (ships, planes) will remain without cosmic navigation and ground support. In these situations, the military XNAV pulsar navigation system will not be available for civilian navigation. But even in the absence of any threats to space and ground navigation and communication systems of the fourth and fifth generation, the presence of a pulsar navigation system will be an additional source of navigational information in certain areas of the Earth’s surface.

For these two reasons, the development of a terrestrial radio pulsar navigation system for transportation is a very current scientific and applied task that awaits its solution.

In this article, a suitable source of space radio signals—pulsars—is chosen for terrestrial navigation. It is currently the only source of radio signals in space used for spacecraft navigation. Incorporating such a coarse pulsar navigation system into the currently existing high-precision multi-sensor navigation system for monitoring and control could be very important.

The passive pulsar radio navigation system could be used on Earth as an additional, increasing credibility and reliability in the conditions of various interferences, which works together with other GNSS radio navigation systems and radio communication technologies based on 4 and 5G generation. 

The main problem arising in the current use of passive terrestrial navigation of transport with space emitters, pulsars, or other space bodies is technological. It is related to the size of the antenna mirrors, the lack of cheap and very sensitive receiving devices, as well as the lack of a developed theoretical basis for their creation. It should be noted that the advancement of communication technologies and their technological development as well as their realization for wide consumption, led to the emergence of a new generation of radio amateurs/radio astronomers [33,34]

The paper shows that in theoretical papers on space and aviation navigation published years ago, systems with small receiving antennas (10 sq.m) using pulsar signals have rather coarse positioning accuracy [2,21,23,26,27,31,32]. 

Like GNSS systems, they use signals from 4 suitable pulsars (out of 50) to position the object. These pulsars are not uniformly distributed but are grouped in certain directions (at least six clusters can be determined). The best pulsars for solving the problem of positioning (the ambiguity of resolution) are pulsars with large repetition periods of the emitted pulses. It turns out that they are also the most powerful pulsars in terms of the SNR of the emitted pulses. Such a pulsar is the pulsar B0329+54 researched in the article. Theoretical estimates of positioning accuracy of space objects previously obtained by Sala’s team [2] are as follows: 0.05 ∙ 10^6^ m—for 10 min of observation; 0.3 ∙ 10^5^ m (50–30 km)—for 100 min of observation; 0.1 ∙ 10^5^ m—for 400 min of observation. Under all other ideal operating conditions of the space navigation system, these values of positioning accuracy are grossly inappropriate for real-time operation. 

In 2013, within the European project under the 7th Framework Program “Pulsar Plane” the issues of creating an aviation radio navigation system working with signals from radio pulsars were investigated [21,22,23,24,25,26,27]. As a result of the work on the project, experimental [23] and theoretical [26,27] estimates of the accuracy of the time of delay (TOA) of signals obtained by different methods have been obtained. The articles [21,23] propose real (experimental) rough position estimates (in time only) of the TOA obtained using signal records from pulsar B0329+54. These recordings were made by the radio telescopes with antenna sizes (500–7000 m^2^) at the Westerbork and Dwingeloo observatories in the Netherlands, (Figure 9 and Figure 10). The TOA estimates presented in the paper were obtained at high SNR values (20–30 dB). They are in the range from 24 km to 1 km with an observation time of 2–3 min. They approach the CRLB estimates, which for the pulsar B0329+54 are from 1 to 10 nm. These estimates were calculated after signal and statistical processing in the MATLAB environment. This is grossly inadequate for aircraft navigation and is perhaps more suitable for marine navigation.

The radio telescope in Dwingeloo uses a single antenna with an area of 500 m^2^ (with gain G [dB] = 16.99 dB) while the radio telescope Westerbork uses 14 such antennas, which correspond to an antenna with an area of 6000 m^2^ (with gain G [dB] = 27.78 dB). The real data from Westerbork contains the noisy complex baseband signal sampled at a frequency of 40 MHz. The frequency band of the input signal is 20 MHz with 28,582,316 samples. The experimental data from Dwingeloo are sampled at a frequency of f_S_ = 70 MHz, and the total number of samples of the input data within a single repetition period is N = 499,800,000.

In [21,26,27], when calculating the theoretical CRLB estimates of the TOA, a methodology of Sala was used, which takes into account the parameters of the signals from three pulsars (from the EPN pulsar database), the receiver parameters to determine the output signal-to-noise ratio (antenna of 10 sq.m, system temperature T = 15–150 K), and also signal processing with accumulation [23]. The results obtained in [26,27] show that the accuracy of distance determination in the aviation navigation system using signals from five pulsars is of the order of tens of kilometers. For system temperature T = 15–150 K accuracy 0.2 ∙ 10^5^ m—for 35 min of observation, and 0.25 ∙ 10^5^ m—for 865 min of observation. This is grossly inadequate for aircraft navigation and is perhaps more suitable for marine navigation. 

## 4. Methodology for Determining the TOA Accuracy in Transport Navigation with Pulsar Radio Signals 

**In Section 4**, one of the methodologies for calculating a theoretical CRLB estimate of the TOA accuracy proposed by Sala in [2] is used. According to this methodology, a theoretical CRLB estimate of the TOA accuracy depends on the parameters of the pulsar signal profile and the SNR value at the meter input. This approach is necessary to compare experimental estimates (coarse and CRLB) of the TOA accuracy depending on the SNR level. According to [2], the main problem in navigation with small antennas using several pulsar signals is the impossibility of accumulating such a level of SNP in the extracted pulsar signal that is sufficient to estimate the TOA with the required accuracy. This paper compares the obtained experimental CRLB estimates of the TOA accuracy with the rough experimental estimates given in [23,24,25]. In both types of experimental estimates (coarse and CRLB) of the TOA accuracy used the signal-to-noise ratio, which has been estimated from recordings of real signals from the pulsar B0329+54 made by the radio telescopes Dwingeloo and Westerbork in the Netherlands. The dependence of the obtained experimental estimates of the TOA accuracy on the real SNR level at the output of the receiver in the radio telescope was also investigated.

### 4.1. Theoretical Estimation of the TOA Accuracy in a Pulsar Navigation System

The accuracy of the TOA estimation (*τ_i_*) in radio pulsar navigation can be theoretically assessed using the following parameters: (i)—receiver parameters and (ii)—pulsar parameters. It is assessed by means of the Cramer–Rao low bound (*CRLB*) on the TOA estimate variance *σ^2^(τ_i_)*, given as:(1)$$\sigma^{2}\left(\tau_{i}\right)\geq CRLB\left(\tau_{i}\right)=\frac{1}{{SNR}_{i}^{2}BT_{int}}C\left(i\right)=\frac{1}{{SNR}_{i,L}^{2}}C\left(i\right)$$

In (1), *SNR_i_* is the signal-to-noise ratio (*SNR*) when detecting the single pulse from the i-th radio pulsar, *T_int_* is the receiver integration time, which is related to the number of processed samples *L* through the receiver bandwidth as:(2)$$L=BT_{int}$$

The parameter *C(i)* in (1) depends only on the pulsar mean power profile shape and can be calculated in two ways (discrete Fourier transform or approximation) of the pulsar mean power profile shape *p*^2^(*nT_s_*), *n* = −*N_T_*/2, …, *N_T_*/2^−1^ (available in EPN pulsar database).

#### 4.1.1. Fourier Transform of p^2^(nT_s_)

In [26,27], it is supposed to calculate the parameter *C(i)* as:(3)$$C(i)=\frac{1}{{\left(2\pi \right)}^{2}Q_{timing}(i)}$$
where the parameter *Q_timing_(i)* is:(4)$$Q_{timing}\left(i\right)=\sum_{k=-N_{T/2}}^{N_{T/2}-1}{\left|\frac{k}{T_{s}N_{T}}\frac{{∆}_{i}(k)}{{∆}_{i}(0)}\right|}^{2}$$
where ${∆}_{i}\left(k\right)=\sum_{n=N_{T/2}}^{N_{T/2}-1}p^{2}(nT_{s})e^{-j\frac{2\pi }{N_{T}}kn},k=-\frac{N_{T}}{2},\ldots ,\frac{N_{T}}{2}-1$.

In (4), ∆*_i_(k)* is the discrete Fourier transform of the pulsar mean power profile shape *p^2^*.

#### 4.1.2. Approximation of p^2^(nT_s_)

The pulsar mean power profile *p^2^(nT_s_)* can be approximated in term of the pulsar pulse parameters that can be taken from the available pulsar database: the pulse duration at 10% (*T_10_*) and 50% (*T_50_*) the peak intensity. According to [2], the parameter *C(i)* in (1) can be calculated as:(5)$$C\left(i\right)=\frac{1}{2}\frac{T_{p}^{2}}{T}T_{ef}\frac{p_{p}^{4}}{S_{1}^{2}}$$

The parameter *T*_p_ in (5) is the complete pulsar pulse duration can be calculated using the pulse profile parameters *T*_10_ and *T*_50_:(6)$$T_{p}=\frac{1}{4}(5T_{10}-T_{50})$$

The last two terms in (5), can be determined by the parameters *T*_1_ and *T*_2_, which in turn are also can be expressed through the parameters *T*_10_ and *T*_50_:(7)$$T_{ef}=\frac{T_{1}T_{2}}{T_{1}+T_{2}}\;\;\mathrm{and}\;\;\;\;\;\frac{p_{p}^{4}}{S_{1}^{2}}={\left(\frac{T_{1}+3T_{2}}{2T_{1}+2T_{2}}\right)}^{2}$$

In (7), the parameters *T_1_* and *T_2_* are calculated as: (8)$$T_{1}=\frac{5}{8}(T_{10}-T_{50})\;\mathrm{and}\;T_{2}=\frac{1}{2}T_{50}$$

### 4.2. Experimental Estimation of the TOA Accuracy Using Pulsar Signal Records

The expression in the denominator of (1) is the squared *SNR* when detecting using accumulation of *L* pulses from the *i*-th radio pulsar, that is:(9)$${SNR}_{i}^{2}L={\left({SNR}_{i}\sqrt{L}\right)}^{2}={SNR}_{i,L}^{2}$$

When conducting experiments, it is possible to record signals from a pulsar. Then, by processing the signal, you can obtain an estimate of the parameter *SNR_i,L_* in (9) and use this estimate to calculate both a rough estimate and a more accurate estimate (*CRLB*) of the TOA measurement.

#### 4.2.1. Rough Estimation

The rough estimate of the TOA variance *σ^2^(τ_i_)* can be approximately calculated as a result of the processing of the records of *L* pulses received from a pulsar during the experiment. As a result, it can be obtained the estimates of *SNR* ($\hat{{SNR}_{i,L}}$) and the width of the cross-correlator output ($\hat{W_{L}}$) at 50% the peak level. Then, the rough estimate of the *TOA* variance *σ^2^(τ_i_)* can be approximately assessed as:(10)$$\hat{\sigma^{2}}\left(\tau_{i}\right)={\left(\frac{\hat{W_{L}}}{\hat{{SNR}_{i,L}}}\right)}^{2}$$

#### 4.2.2. CRLB Estimation

The *CRLB* estimate of the *TOA* variance *σ^2^(τ_i_)* can be found by using the estimate of *SNR* ($\hat{{SNR}_{i,L}}$) in the expression (1). This estimate is calculated as:(11)$${\hat{\sigma }}^{2}\left(\tau_{i}\right)\geq CRLB\left(\tau_{i}\right)=\frac{1}{{\hat{SNR}}_{I,L}^{2}}C\left(i\right)$$

## 5. Numerical Results for TOA Accuracy Estimates after Processing Pulsar Signals from the Pulsar B0329+54

Signals from pulsar B0329+54 received by radio telescopes at Dwingeloo and Westerbork observatories, before recording pass through the following physical devices: antenna, antenna-feeder device, high and intermediate frequency amplifiers, intermediate frequency filters, and ADC). This means that the digital recordings of pulsar signals that are used to estimate the TOA account for all possible signal losses in a real navigation system.

Therefore, the experimental TOA estimates obtained in the paper are realistic estimates because they are based on real pulsar signals in a specific radio telescope. These experimental evaluations realistically reflect the quality of the timing in a radio navigation system using radio telescopes at Dwingeloo and Westerbork observatories.

**In Section 5**, experimental CRLB estimates of the TOA accuracy (time and distance) obtained by Sala’s methodology and rough estimates of the TOA accuracy obtained by methodology from [1] are presented. Both types of estimates have been calculated as a function of the SNR level experimentally estimated at the meter input by using the real signals from the pulsar B0329+54.

This allows a comparison to be made between the two types of experimental estimates (rough and CRLB) of the TOA accuracy at the same magnitudes of the measured signal-to-noise ratio. The obtained results show that the difference between the two estimations of TOA accuracy (coarse and CRLB) decreases with the growth of SNR at the input of the meter. The obtained results show at what SNR level at the input of the meter can be expected measurement errors in the TOA and distance, which are acceptable in a practical navigation system.

In Table 1 and Table 2, the experimental CRLB estimates of the TOA accuracy are calculated using Sala’s methodology according to (11) where SNR has been estimated by processing the signals from the pulsar B0329+54. 

The rough experimental estimates of the TOA accuracy given in Table 1 and Table 2 are obtained by the methodology from [1], where the SNR values have been estimated by processing the signals from the same pulsar and the same radio telescopes. These rough estimates of the TOA accuracy have been earlier published in the papers [23,24,25,26]. In the papers [10,11,12], when processing experimental records of the pulsar B0329+54, two algorithms were used for the rough estimation of TOA. One of them used only folding (synchronous pulse accumulation) evaluating the effectiveness of the algorithm depending on the number of accumulated pulses. The second algorithm used a post-folding matched filter whose impulse response was matched to the signal profile. Figure 11 shows the strong output signal after folding over 125 periods, obtained in Westerbork.

In this figure it is clear to see the peak of the accumulated pulse signal against the noise background. Next, the result of both operations, the folding (synchronous accumulation) and matched filtering, of the same signal (real, imaginary components and power) obtained at the Westerbork Observatory is shown in Figure 12. From this figure can be seen that after this processing the pulsar signal peak with its characteristic two side peaks is clearly visible and also can be seen that the output noise is suppressed. This shows the effectiveness of the proposed signal processing approach (folding and matched filter) in processing signals from pulsar B0329+54 [23,24,25]. Table 1 and Table 2 present TOA estimates obtained in [23,24,25] when processing experimental data from the pulsar B0329+54 when these two types of signal processes were used.

The accuracy of the rough experimental estimate of TOA was obtained in [16], according to the following expression:(12)$$\sigma_{TOA}=∆P/({SNR}_{cor}N)$$

In (12), Δ*P* is the width of the normalized pulsar profile determined at the level of 0.7 (number of samples), *P* is the repetition period of pulsar signals (seconds), *SNR_cor_* is the value of SNR determined at the cross-correlator input, and N is the total number of samples within a repetition period. It can be seen from (12) that the estimate calculated in this way depends on the level at which the width of the pulsar pulse profile is determined. This estimate would be further improved if the pulse profile width at the level 0.9 was used.

In [23,24,25,26], when processing the experimental recordings of signals from B0329+54, all samples in the receiver bandwidth were used for TOA estimation, assuming that these are independent samples. However, after passing white Gaussian noise through a linear receiver system with limited frequency bandwidth, it is known to be correlated. The sampling interval of the signal in this case must be greater than the correlation time of the noise. Therefore, in this case, it is necessary to perform downsampling of the samples in the frequency band of the receiver. In the experimental records of pulsar signals obtained from radio telescopes, there is no information about the limits of the repetition periods of pulsar pulses. Therefore, a special organization of data reading is required to ensure high folding quality. In [23,24,25], no special studies were conducted in processing the signal records in order to improve the folding quality. This partially explains the discrepancies between theoretical and numerical results of the TOA estimates accuracy (rough and CRLB) obtained in this paragraph depending on the CNR level (number of pulses).

Table 1 and Table 2 present the experimental estimates of the TOA accuracy (rough and CRLB) for various SNR values (or the number of periods Np used to accumulate the signal) obtained by processing signals from the pulsar B0329+54 at the radio observatories Westerbork and Dwingeloo in the Netherlands.

The results in Table 1 and Table 2 show that the two types of experimental TOA accuracy estimates (rough and CRLB) obtained for pulsar B0329+54 strongly depend on the SNR level at the meter input. The rough estimate of the TOA accuracy lags behind the CRLB estimate by at most an order of magnitude. The accuracy of the two types of experimental estimates of TOA (rough and CRLB) depends on a number of real losses in SNR: in signal processing in the antenna feeder system, in the high-frequency and intermediate-frequency part of the receiver, in the ADC and other equipment, used in the Dwingeloo and Westerbork radio telescopes. 

Real estimates, or experimental rough estimates of the TOA accuracy (according to the results presented in Table 1 and Table 2), were obtained for the observed pulsar B0329+54, from the radio equipment of the radio telescopes at the Westerbork and Dwingeloo observatories in the Netherlands.

They are within the limits of ~24 km to ~1 km (2.3805 · 10^4^–0.9405 · 10^3^ m) and were obtained during an observation time of about 2–3 min, to accumulate sufficient energy to obtain SNR ratios of 20–30 dB for TOA estimation. The observation time according to Table 1 and Table 2 is determined by the number of foldings of the pulsar signals (125–150), as the repetition period of the pulsar B0329+54 signal is T = 0.71452 s.

The experimental rough estimates of the TOA accuracy (real estimation) presented in this paper (in the same Table 1 and Table 2) are similar to the theoretical CRLB estimates and are within the limits of one kilometer (0.3132 · 10^4^–0.2909 · 10^3^ m). The theoretical CRLB estimates are the limits of obtaining TOA accuracy, which cannot be reached in real navigation systems.

Table 3 presents CRLB estimates of TOA accuracy calculated by Sala’s method according to expression (1) depending on the SNR values, the corresponding observation time of the pulsar B0329+54, for the case of a small antenna (10 sq.m) suitable for the use in a navigation system of transport vehicles.

The CRLB estimates of TOA presented in Table 3 show that using the small antenna (10 sq.m) and operating in the real-time mode (1–5 min) the rough estimate of the error in the distance (from TOA) is of about 31,782 m. 

The obtained rough experimental and theoretical estimates of TOA for positioning and navigation using the signal from pulsar B0329+54, for the navigation pulsar system presented in the paper (at different antenna sizes 500 m^2^ and 7000 m^2^), is comparable to the accuracy of other navigation systems on earth: INS (Typical Inertial Navigation Systems) and Omega/VLF and Loran, several nautical miles (1 to 10).

The article states that it is necessary to look for opportunities to significantly reduce the size of the antenna system compared to those used in the experiment (500 m^2^ and 7000 m^2^) while ensuring a high level of SNR. The paper also suggests all the different approaches proposed in statistical radar theory [30,35] to increase the receiver output SNR ratio while saving antenna dimensions.

## 6. Functional Structure of Signal Processing in a Pulsar Transport Navigation System

The article is devoted to the theoretical aspects regarding the development of an autonomous transport radio navigation system with a small receiving antenna, using radio signals from pulsars, similar to navigation systems for space navigation. Therefore, the proposed structure of a transport navigation system is similar to that of a spacecraft. It should consist of input (X-ray and radio pulsar signals), three main blocks—instruments, timing estimation, and location algorithm [2]—and output (position estimates). In this paper, only the issues of processing pulsar signals are considered, and that is related to the input and the first two blocks of the navigation system. This is also evident from the proposed functional structure of signal processing in a pulsar transport navigation system presented in Figure 13. 

In accordance with [2], we propose that in a transport navigation system, the antenna system should synthesize beams directed at different directions, where the used radio pulsars are located. Zeros are formed in the diagram of reception of the used pulsar signals in order to remove noise sources such as the sun. That requires an antenna geometry based on single antennas, each with its own receiver, and processing the received signal from the observed pulsar [2].

On Earth’s surface, only radio pulsars can be used for transport navigation. Like GNSS systems, it uses signals from 4 suitable pulsars (out of 50) to position the object. These pulsars are not uniformly distributed but are grouped in certain directions (at least six clusters can be determined) [2]. 

For transport navigation, more powerful second pulsars can be used, which can be observed with smaller antennas. These pulsars have a pulse emission period of the order of 1 sec (second pulsars). The TOA measurement error for these pulsars is of the order of tens–hundreds of microseconds. For millisecond pulsars, the pulse repetition period is in the order of several milliseconds. These pulsars have less radiation power and therefore require antennas with a larger area for observation, and the measurement error of TOA is less than 10 microseconds [31,32].

Geometric reduction of accuracy in pulsar navigation requires that the positioning reference sources are widely distributed in angles of arrival. If the observed group of pulsars is large, better accuracy characteristics will be achieved (in a single-channel and single-antenna observation). But at the same time, the minimum integration time for each individual window will increase, along with the minimum time for obtaining an estimate of the first position. Therefore, when choosing the pulsars for observation, there should be trade-offs whether to select a group of pulsars achieving higher positioning accuracy with sufficient time for integration or a group of pulsars requiring less time for observation with lower accuracy.

The strategy of the transport navigation algorithm is first to group several pulsars (four) in order to use small accumulation time and obtain an average position accuracy, and then to include more pulsars for improving the position accuracy.

As can be seen from the proposed functional scheme for processing pulsar signals in the transport navigation system (shown in Figure 13), in the channel of each different pulsar, firstly, a space–time processing of the incoming signal is carried out in the antenna array, as in GNSS systems [36]. 

For each pulsar, in each antenna array element, synchronous pulse accumulation of the incoming pulse trains is first performed to improve the SNR at the output of the array element. We assume that the number of pulses for the synchronous accumulation for each different pulsar is predetermined and considered fixed. After the synchronous accumulation (folding) of the pulses, the summation of the output signals from all elements of the antenna array (spatial channels) follows to obtain the total single pulse (quasi-deterministic signal) with an additional increased SNR. Multi-channel processing of quasi-deterministic signals at the output of the antenna array for estimation of their unknown information parameters is carried out sequentially (for each information parameter) or jointly (i.e., simultaneously). 

In the proposed paper, the management and formation of the beams in the antenna array is not considered, but it is only assumed that for each particular pulsar, there is a specific direction of observation in space. The antenna array and timing algorithm proposed in the article, using the rough estimate of the TOA accuracy, is similar to the algorithm for processing GPS signals in transport navigation GNSS systems [36]. But unlike GNSS satellites, which move much faster than vehicles, pulsars are considered stationary relative to the vehicle. 

## 7. Optimal Multichannel Space–Time Algorithm for Estimation of TOA and Doppler Frequency of Vector Pulsar Signals

The main scientific task that is solved in the paper is the adaptation of all the possible approaches and algorithms suggested in the statistical theory of radars in the suggested signal algorithm for antenna processing and to evaluate the parameters of the TOA and DS pulsar signals, in order to increase the SNR ratio at the receiver output, while preserving the dimensions of the antenna.

In this chapter, we tried to apply a mathematical apparatus for vector description of signals to pulsar signals received at the input of a linear antenna array. The vector description of the received signal allowed us to describe the spatio-temporal processing of the pulsar signal in the linear antenna array. 

First, in each element of the antenna array, the accumulation (folding) of all pulsar pulses received at the input of the antenna array is performed. The next stage of processing is the summation of the accumulated signal on all elements of the linear antenna array. Furthermore, the SNR can also be increased by receiving broadband pulsar signals at several different observation frequencies, since pulsars are known to emit signals over a very wide radio frequency range. Therefore, in the same beam of the antenna array of the receiver, a sufficient number of signals from the same pulsar at different observation frequencies can be simultaneously received. 

All these stages of processing allow for a significant increase in the SNR of the accumulated pulse at the input of the pulsar timing TOA estimation, to the needed SNR ratios of 20–30 dB for TOA estimation [23,26,27], depending on the antenna size.

The average profile of pulses emitted by a pulsar is known to be one of the main characteristics of every pulsar represented in the EPN database. Using the fact that the average pulse profile from a pulsar is the average of the pulses in the pulse train, we reformulated the signal model at antenna array output as a single radio pulse with two unknown and non-random parameters (TOA and Doppler frequency) and with two random parameters (amplitude and initial phase). This reformulation of the signal model allowed us to solve the task of joint estimation of the signal parameters at the output of the antenna array using the standard maximum likelihood ratio method.

Non-Bayesian estimation methods are known to be used to estimate unknown non-random parameters. The maximum likelihood ratio method received the greatest distribution. According to this method, the ranges of the evaluated parameters, the Doppler shift of frequency *∆f* and the delay *τ*, are replaced by a finite set of discrete values of the parameters in the corresponding interval [35]. Then the likelihood ratio for the entire estimated interval is represented as a set of likelihood ratios for all values of the parameter in that interval. In such a case, the estimation algorithm is multi-channel, with each channel forming a conditional likelihood ratio. The likelihood ratio is the ratio of the conditional probability density in the presence of the estimated parameter to the conditional probability density in the absence of the estimated parameter. In the presence of random non-informative parameters (in this case *a* and *φ*_0_), the likelihood ratio is averaged over these parameters using their probability distributions. The maximum likelihood ratio is reached at parameter values that are assumed to be the maximum likelihood estimates of those parameters. These multichannel estimates are approximately equal to the joint maximum likelihood estimates of TOA and Doppler frequency shift, and the degree of approximation depends on the selected number of channels. 

This signal processing can be implemented as a correlation or filter [35,37,38,39]. In this paper, the case of joint multi-channel correlation processing is considered. Unlike the filter implementation of the estimation algorithm, the parallel implementation of the joint correlation measurement of the two unknown information parameters of the quasi-deterministic signal (TOA and Doppler frequency shift) requires a larger number of measurement channels but reduces processing time. Multichannel measurement of an unknown signal information parameter according to the method of maximum likelihood ratio implies the division of the assumed interval of variation of the parameter into measurement channels [35,37,38,39]. Only the channel that provides the maximum value (amplitude) of the likelihood ratio of the output is selected. The accuracy of the obtained estimates depends on the selected number of channels. In each measurement channel of the Doppler frequency shift (from the first to the last), parallel correlation processing is performed between a quasi-deterministic total signal at the output of the antenna array and the reference signal with all fixed values of the informative parameter TOA from its measurement range. At their output, as can be seen from Figure 13, there is a scheme for choosing the maximum amplitude of the decisive statistics. Only that measurement channel “Doppler frequency shift—TOA” is selected, which is maximal in amplitude from all possible channels.

The signal processing algorithm shown in Figure 14 can be described mathematically as follows [35,37,38,39]. Let us assume that for the observation time *T_obs_*, a sequence of *N_p_* high-frequency pulses emitted by a pulsar with the central frequency *f*_0_ and the pulse repetition period *P* arrives at the input of each element of a *K*-element antenna array. We also assume that each pulse in the pulse sequence has two unknown informative non-random parameters, delay time (*τ*) and frequency Doppler shift (∆*f*) to be estimated, and has a random amplitude shape (*p*) and a random initial phase (*φ*_0_). At the output of an antenna array, the signal matrix y(t) of size (*N_P_* × *K*) is formed.
(13)$$y(t)=\left[x_{1}\left(t-\tau \right)x_{2}\left(t-\tau \right)\ldots \ldots .x_{K}(t-\tau )\right]+{\left[\eta \right]}_{K}$$
where *x_k_(t − τ)* is a signal vector of size (*N_P_* × 1), representing a sequence of *N_P_* pulses arriving with a delay *τ* at the input of the *k*-th element of the antenna array (*k* = 1 ÷ *K*), [*η*]*_K_* is the matrix of size (*N_P_* × *K*) of Gaussian noise in the antenna array. Each element of the input signal matrix *x(t − τ)* of size (*N_P_* × *K*), i.e., *x*_nk_*(t − τ)* (*n* = 1 ÷ *N_P_*, *k* = 1 ÷ *K*) represents the *n*-th pulse from the pulse sequence received at the input of the *k*-th element of the antenna array. This impulse is mathematically described as:(14)$$x_{nk}(t-\tau )=p_{n}(t-\tau )cos\{2\pi \left(f_{0}-∆f\right)\left[t-\left(n-1\right)P-\tau \right]+\varphi_{k}+\varphi_{0}\}$$

In (14), *∆f* is the Doppler frequency shift depending on the speed of the transport vehicle, *φ_k_* is the phase difference in the *k*-th antenna array element, *φ_0_* is a random initial phase, and *p_n_(t)* is a random shape of the *n*-th pulse determined in the interval *t* = [0, *P*] [35,37,38,39]. The complex envelope of the signal matrix *y(t*) at the output of the antenna array has the form:(15)$$Y\left(t\right)=\left[X_{1}\left(t-\tau \right)X_{2}\left(t-\tau \right)\ldots \ldots .X_{K}\left(t-\tau \right)\right]+{\left[\overset{´}{\eta }\right]}_{K}$$

In (15), *Y, X* and *ή* are, respectively, the complex envelopes of the output signal, the input signal and the noise of the receiving antenna. Considering (14), the complex envelope of each pulse in the pulse sequence arriving at the input of the k-th element of the antenna array can be written as:(16)$$X_{kn}\left(t-\tau \right)=p_{n}\left(t-\tau \right)\exp\left[-j\phi_{n}\left(f_{0},∆f\right)\right]\exp\left[-j\Psi \left(∆f,\tau \right)\right]\exp\left(-j\varphi_{k}\right)exp(j\varphi_{0})$$

Phases *ϕ_n_ (f_0_,∆f)* and *Ψ(∆f,τ)* are calculated as:(17)$$\phi_{n}\left(f_{0},∆f\right)=2\pi {(f}_{0}-∆f)(n-1)P$$
(18)$$\Psi \left(∆f,\tau \right)=2\pi f_{0}\tau +2\pi ∆f(t-\tau )$$

Since the intermediate frequency *f_0_* (in MHz) is much larger than *∆f* (in KHz), i.e., *f_0_* >> ∆f, then (17) can be simplified:(19)$$\phi_{n}\left(f_{0}\right)=2\pi f_{0}(n-1)P$$

It follows from (19) that the phases ϕn depend only on the intermediate frequency and on the pulse delay relative to the first pulse in the pulse sequence. 

According to the block diagram in Figure 13, the processing of the complex envelope matrix *Y(t)* starts first with the synchronous accumulation of the pulses in each element of the antenna array (the first stage of space–time processing). This processing allows us to increase the SNR of the output signal of each element of the antenna array.

Mathematically, this stage of space–time processing is described as the multiplication of the phase vector that compensates phases *ϕ_n_* in the complex envelope vector *Y(t)*:(20)$$\left[Y_{1}^{\prime },Y_{2}^{\prime },\ldots ,Y_{K}^{\prime }\right]=\left[e^{\phi_{1}},e^{\phi_{1}},\ldots e^{\phi_{1_{Np}}}\right]Y(t-\tau )$$

As a result of this operation, at the output of each *k*-th element of the antenna array, a pulse is formed with a complex envelope, which is a sum of the complex envelopes of all pulses of the pulse sequence [35,37,38,39]:(21)$${Y\prime }_{k}=\sum_{n=1}^{N_{P}}p_{n}\left(t-\tau \right)\exp\left[-j\Psi \left(∆f,\tau \right)\right]\exp\left(-j\varphi_{k}\right)\exp\left(j\varphi_{0}\right)+\eta_{k}\prime$$

Physically, this is implemented using a transversal filter with (*N_P_* − 1) delay lines in each element of the antenna array. Since the SNR of the individual pulse in the pulse sequence is extremely low (<−50 dB), the purpose of this preliminary synchronous accumulation of the received pulses (folding) in each element of the antenna array is to increase the SNR of the accumulated pulse by summing the powers of all incoming pulses.

The next stage of space–time signal processing is realized by the weighted summation of signals from each element of the antenna array with the corresponding phase weighting coefficients [35,37,38,39]. These weighting coefficients (phase differences) compensate for mutual phase shifts, i.e., *φ_k_*, in the elements of the antenna array. As a result, the beam of the antenna is pointed in the direction of the pulsar and the received signal is maximal in that direction because it is a sum of signals from all antenna elements.

For simplicity, we assume that the pulse delay *Y’_k_* in the various elements of the antenna array can be neglected. The phase vector of the antenna array, i.e., [*exp*(−*φ_k_*)], *k* = 1…*K*, does not depend on time, but only on the position of the element in the antenna array with respect to the first element. The phases *φ_k_* can be compensated if the vector (21) is scalar multiplied by the vector [*exp*(*φ_k_*)], *k* = 1…*K*. Mathematically, this is described as:(22)$$Y_{\Sigma }=[Y_{1}^{\prime },Y_{2}^{\prime },\ldots ,Y_{K}^{\prime }]{\left[e^{j\varphi_{1}},\ldots {,e}^{j\varphi_{K}}\right]}^{T}$$

As a result of operation (21), a scalar (summed) signal with the following complex envelope is obtained:(23)$$Y_{\Sigma }=K\exp\left[-j\Psi \left(∆f,\tau \right)\right]\exp\left(j\varphi_{0}\right)\sum_{n=1}^{N_{p}}p_{n}(t)+\eta_{n}\prime \prime$$

It follows from (23) that the second stage of space–time processing further increased the SNR of the signal by *K* times in result of summing signals from all elements of the antenna array. Separating the real part of the complex envelope (23) gives the summed signal:(24)$$y_{\Sigma }\left(t\right)=K\sum_{n=1}^{N_{p}}p_{n}\left(t-\tau \right)cos[\Psi \left(∆f,\tau \right)+\varphi_{0}]$$

It is known from the literature that the average pulse profile of a pulsar, *p(t)* can be roughly defined as the mathematical expectation of the pulse amplitudes in the received pulse sequence [30]:(25)$$p\left(t-\tau \right)=\frac{1}{N_{P}}\sum_{n=1}^{N_{P}}p_{n}(t-\tau )$$

Considering (25), the signal (24) can be represented as:(26)$$y_{\Sigma }\left(t\right)=aAp(t-\tau )cos[\Psi \left(∆f,\tau \right)+\varphi_{0}]$$

In (26), the parameter *A* = *KN_P_* accounts for the increase in SNR of the input signal after the two stages of space–time processing. The parameter *a* is a random variable that accounts for the random nature of the pulse amplitudes in the input sequence with unit dispersion. It follows from (26) that the signal *y_Σ_(t)* can be defined as a quasi-deterministic signal with two random non-informative parameters (*a*, *φ_0_*) and two unknown informative non-random parameters (*τ*,*∆f*) that must be estimated.

According to the method for maximum likelihood, the ranges of variation of the Doppler frequency shift *∆f* and the time delay *τ*, are replaced by a set of discrete values from their intervals of variation [35,37,38,39]. If there are random non-informative parameters (in the case of *a* and *φ_0_*), the likelihood ratio is averaged over these parameters.

In our case, the interval of possible values of the Doppler frequency shift [*∆f_min_*, *∆f_max_*] is replaced by a set of its *M* discrete values [*∆f*_1_, *∆f_2_*, … *∆f_i_*, … *∆f_M_*]. The interval of possible values of the delay time (*τ*), i.e., [0, *P*], is also replaced by a set of its discrete values [*τ_1_*, *τ*_2_, …, *τ*_N_]. The number of discrete parameter values in the range of variation depends on the accuracy with which we want to estimate the particular parameter. The conditional likelihood ratio *Λ*(*y_Σ_*|*∆f*,*τ*) is replaced by a discrete set {*Λ*(*y_Σ_*|*∆f_i_*,*τ_j_*), *i* = 1,2,..*M*, *j* = 1,2,..*N*} [25,26,27,28]. As a result, we have a (*M* × *N*)-channel algorithm for the estimation of *∆f* and *τ*. In each (*i,j*)-th channel, the likelihood ratio is formed for the fixed values of the parameters (*∆f_i_*, *τ_j_*). To find the estimate of a pair of parameters (*∆f, τ*) the channel is selected in which the conditional likelihood ratio has a maximum value, i.e., [35]:(27)$${(\hat{∆f,}\tau }_{TOA})=\arg\underset{{∆f}_{i},\tau j}{\max}\{\Lambda (y_{\Sigma }|{{∆f}_{i},\tau }_{j})\}$$

Since the accumulated signal *y_Σ_(t)* has two random parameters *a* and *φ_0_*, the conditional likelihood ratio *Λ(y_Σ_|∆fi,τj)* in the (*i, j*)-channel must be averaged over random uninformative parameters:(28)$$\Lambda \left(y_{\Sigma }|∆f_{i},\tau_{j}\right)=\int_{0}^{\infty }\int_{0}^{2\pi }{\Lambda (y}_{\Sigma }\left|∆f_{i},\tau_{j},a,\varphi_{0}\right)f(a,\varphi_{0})dad\varphi_{0}$$

We assume that the random parameter *a* is distributed according to Rayleigh’s law with unit variance, and the random initial phase *φ_0_* is uniformly distributed in the interval [0, 2π]. For independent random variables *a* and *φ_0_*, the joint density distribution is:(29)$$f\left(a,\varphi_{0}\right)=\frac{1}{2\pi }\left(\frac{a}{\sigma^{2}}\right)exp(-a^{2}/(2\sigma^{2})$$

After replacing *f*(*a, φ_0_*) in (28) by (29), respectively, the conditional likelihood ratio (28) takes the following form according to [25]:(30)$$\Lambda \left(y_{\Sigma }|∆f_{i},\tau_{j}\right)=\frac{N_{0}}{E+N_{0}}exp\left(\frac{{2\sigma^{2}Z}^{2}(∆f_{i},\tau_{j})}{N_{0}(E+N_{0})}\right)$$

The parameter *E* = 2*E*_p_
*σ*^2^ is the average energy of the signal, *E_P_* is the energy of the pulse profile *p(t)*, i.e., *E_p_* = ∫*p*^2^ (*t*)*dt* and *N_0_* is the receiver noise energy. The sufficient statistics *Z*(*∆f_i_,τ_j_*) in (30) is the envelope of the correlation integral, for the case of the random phase and amplitude [35]:(31)$$Z\left(∆f_{i},\tau_{j}\right)=\sqrt{Z_{1}^{2}\left(∆f_{i},\tau_{j}\right)+Z_{2}^{2}\left(∆f_{i},\tau_{j}\right)}$$

The quadrature components of the correlation integral are:(32)$$Z_{1}\left(∆f_{i},\tau_{j}\right)=A\int_{0}^{P}y_{\Sigma }\left(t\right)p\left(t-\tau \right)cos\left[\Psi \left(∆f_{i},\tau_{j}\right)\right]dt$$
(33)$$Z_{2}\left(∆f_{i},\tau_{j}\right)=A\int_{0}^{P}y_{\Sigma }\left(t\right)p\left(t-\tau \right)sin\left[\Psi \left(∆f_{i},\tau_{j}\right)\right]dt$$

It follows from (30) that the algorithm (27) for estimating the Doppler frequency shift and TOA is transformed into the following algorithm:(34)$${(\hat{∆f,}\tau }_{TOA})=\arg\underset{{∆f}_{i},\tau j}{\max}\{Z^{2}({{∆f}_{i},\tau }_{j})\}$$

According to (31), in each (*i,j*)-th channel of the time processing for all values of *∆f_i_* (*I* = 1,...*M*) and *τ*_j_ (*j* = 1,...*N*) the square of the correlation integral *Z*(*τ_i_, ∆f_j_*) is calculated and the channel with this maximum value determines the estimate ${(\hat{∆f,}\tau }_{TOA})$. The structural diagram of this estimation algorithm is shown in Figure 14.

The mathematical apparatus suggested below could be used for the theoretical evaluation of the SNR ratio for TOA estimation, depending on the antenna size, the pulsar signal parameters, and the specificity of all stages of signal processing.

It is known that the accuracy of the CRLB estimator depends on the number of independent signal discrete (*L*) processed during the observation time *T_obs_* = *PN_p_*. After space–time processing of the signal at the output of the *K*-element antenna array (single antenna linear array) the number of processed samples *L* through the receiver bandwidth *B* is:(35)$$L=BT_{obs}=KBPN_{p}$$

It follows from (35) that the number of processed independent samples required to estimate the parameters *∆f* and *τ* can be achieved either by increasing the observation time of the pulsar signal (*T_obs_*), i.e., the number of accumulated pulses *N_p_,* or by increasing the frequency band of the receiver (*B*), or by increasing the number of elements in the antenna array (*K*). The SNR of the accumulated pulse signal obtained after spatial–temporal processing (according to Figure 13) can be additionally increased by using several numbers of linear antenna arrays (*Q*) and receiving the pulsar signals simultaneously at several numbers of observation frequencies (*f_obs_*). Then, the expression (35) will take the form:(36)$$L=BT_{obs}=KBPN_{p}Q$$
where *Q*- is the number of linear antenna arrays. It should be noted that the signal-to-noise ratio of the single pulse received at different frequencies of observation depends on the particular frequency of observation:(37)SNR1(fobs)=Pimp(fobs)Pn

In (37), *SNR*_1_ is the signal-to-noise ratio of the single pulse emitted by the pulsar at the frequency of observation *f_obs_*, *P_imp_*(*f_obs_*) and *P_n_* are the pulse power and the receiver noise power, respectively. Then the accumulated signal-to-noise ratio received at *F_P_* different frequencies of observation and Q linear antenna arrays is:(38)$${SNR}_{L}={SNR}_{1}L=\sum_{n=1}^{F_{P}}{SNR}_{1}(f_{obs})KBPN_{P}Q$$

Therefore, if the accumulation time of the pulse train exceeds a fixed real-time signal processing time, we can reduce it by using an additional number of parallel receive channels to satisfy the necessary signal-to-noise ratio requirements in real-time operation.

## 8. Conclusions

The article is devoted to the theoretical aspects regarding the development of an autonomous radio navigation system with a small receiving antenna, using radio signals from pulsars, similar to navigation systems for space navigation. On Earth’s surface, only radio pulsars can be used for navigation. The paper shows that in theoretical papers on space and aviation navigation published years ago, systems with small receiving antennas (10 sq.m) using pulsar signals have rather coarse positioning accuracy. Like GNSS systems, they use signals from 4 suitable pulsars (out of 50) to position the object. These pulsars are not uniformly distributed but are grouped in certain directions (at least six clusters can be determined). The best pulsars for solving the problem of positioning (the ambiguity of resolution) are pulsars with large repetition periods of the emitted pulses. It turns out that they are also the most powerful pulsars in terms of the SNR of the emitted pulses. Such a pulsar is the pulsar B0329+54 researched in the article. Theoretical estimates of positioning accuracy of space objects previously obtained by Sala’s team are as follows: 0.05 ∙ 10^6^ m—for 10 min of observation; 0.3 ∙ 10^5^ m. (50–30 km)—for 100 min of observation; 01 ∙ 10^5^ m—for 400 min of observation. Under all other ideal operating conditions of the space navigation system, these values of positioning accuracy are grossly inappropriate for real-time operation. 

When using small antennas (with an area of up to tens of square meters) for pulsar navigation, the energy of the pulsar signals received within a few minutes is extremely insufficient to obtain the required level of SNR at the output of the receiver to form TOA estimation, ensuring positioning accuracy up to tens of kilometers. It follows that according to the theory of space navigation by pulsars, no better positioning accuracy can be expected in pulsar navigation of ground transport when using very small antennas with an area of 10 sq.m. If the size of the receiving antenna is increased to practically acceptable sizes (from tens of square meters to hundreds of square meters), then using only pulse folding, we can increase the SNR by several decibels (5−10 dB), which will reduce the observation time to several tens of minutes but will not improve the positioning accuracy, which will be tens of kilometers. 

The transport navigation concept requires significantly reducing the observation time (within a few minutes), providing the transport positioning accuracy that is close to potential. In this case, more powerful second pulsars can be used, which can be observed with smaller antennas. However, these pulsars have a pulse emission period of the order of 1 s (second pulsars). The TOA measurement error for these pulsars is of the order of tens–hundreds of microseconds. The other group of pulsars is the so-called millisecond pulsars, the pulse repetition period of which is of the order of several milliseconds. However, these pulsars have less radiation power and therefore require antennas with a larger area for observation. However, the measurement error of TOA is less than 10 microseconds.

The first scientific task that is solved in the paper is to study the relationship between the SNR of the receiver output, which depends on the size of the antenna, the type of signal processing, and the magnitude of the TOA accuracy estimate, using the records from B0329+54 pulsar. The second scientific task that is solved in the paper is the adaptation of all the possible approaches and algorithms suggested in the statistical theory of radars in the suggested signal algorithm for antenna processing and to evaluate the parameters of the TOA and DS pulsar signals, in order to increase the SNR ratio at the receiver output, while preserving the dimensions of the antenna.

The article proposes real (experimental) rough position estimates (in time only) of the TOA obtained using signal records from pulsar B0329+54. They approach the CRLB estimates, which for the pulsar B0329+54 are from 1 to 10 nm. The article states that it is necessary to look for opportunities to significantly reduce the size of the antenna system compared to those used in the experiment (500 m^2^ and 7000 m^2^) while ensuring a high level of SNR. 

The paper also suggests all the different approaches proposed in statistical radar theory to increase the receiver output SNR ratio while saving antenna dimensions. These approaches include space–time processing of signals in an antenna array, multi-beam reception of the various broadband pulsar signals for navigation (not less than four), simultaneous reception of signals from one pulsar at different observation frequencies. The space–time processing of the pulse signal for each received beam performs the accumulation of all pulses in each antenna array element and the summation of the accumulated signal over all array elements, followed as well as with coherent filtering of the pulse signal in multi-channel processing for estimation of the TOA and Doppler frequency shift. In our opinion, all these processing steps can significantly increase the SNR of the accumulated pulse to the required levels of 20–30 dB for rough TOA estimation [23,26,27], depending on the chosen antenna size. 

The proposed algorithms for space–time processing of the signal in the antenna array and for multi-channel estimation of the TOA and Doppler frequency shift is suitable for real-time operation, as they can be implemented in the form of parallel computing structures (for example, VLSI). 

The proposed estimates of positioning accuracy (TOA only, no phase) in an autonomous pulsar vehicle navigation system would only be suitable for the navigation of large vehicles (sea, air, or land) that do not require accurate navigation at sea, air, or desert. Large-sized antennas with an area of tens of square meters to hundreds of square meters can be installed in such vehicles.

## Figures and Tables

**Figure 1 sensors-23-07010-f001:**
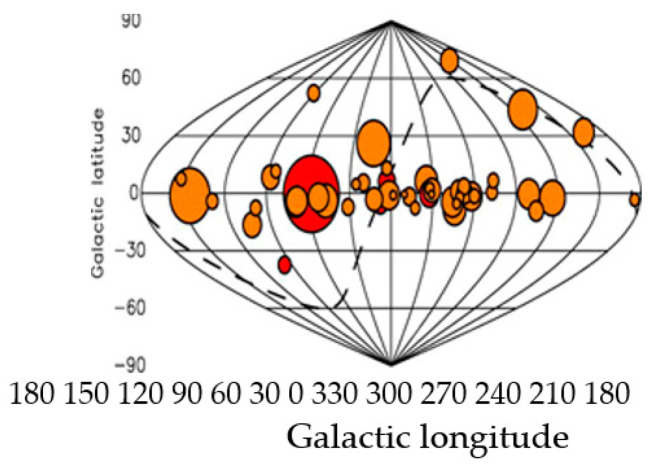
Positions in galactic coordinates.

**Figure 2 sensors-23-07010-f002:**
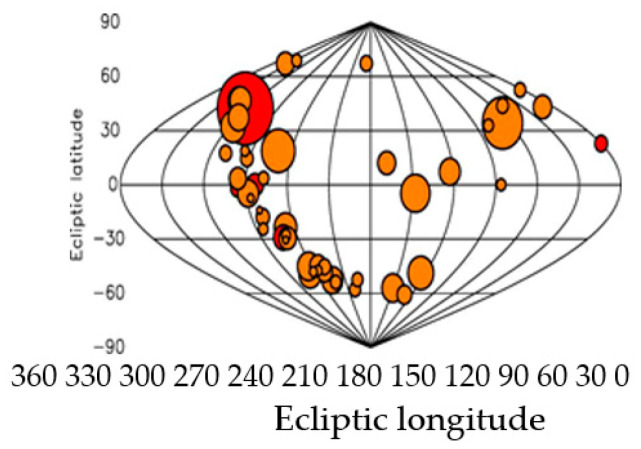
Ecliptic coordinates.

**Figure 3 sensors-23-07010-f003:**
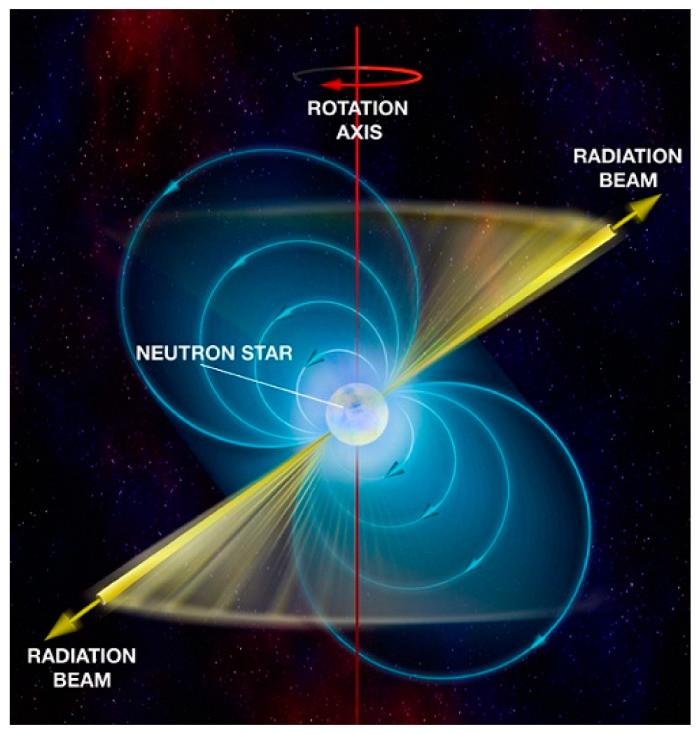
E/m emission from a pulsar.

**Figure 4 sensors-23-07010-f004:**
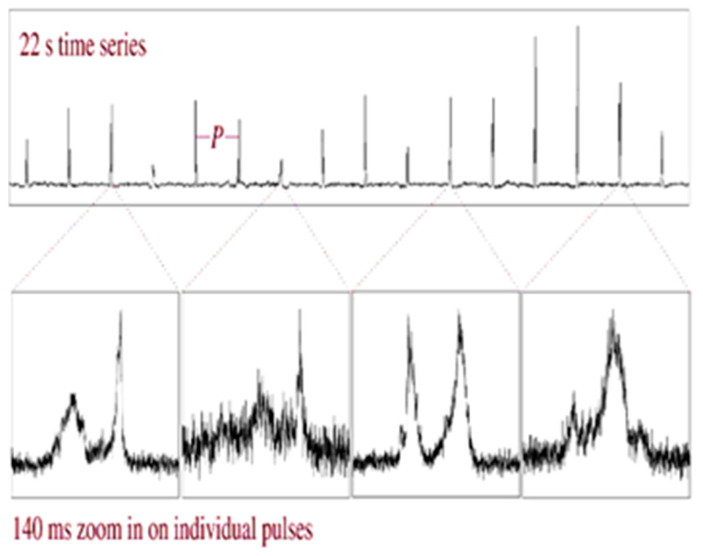
A sequence of pulsar pulses.

**Figure 5 sensors-23-07010-f005:**
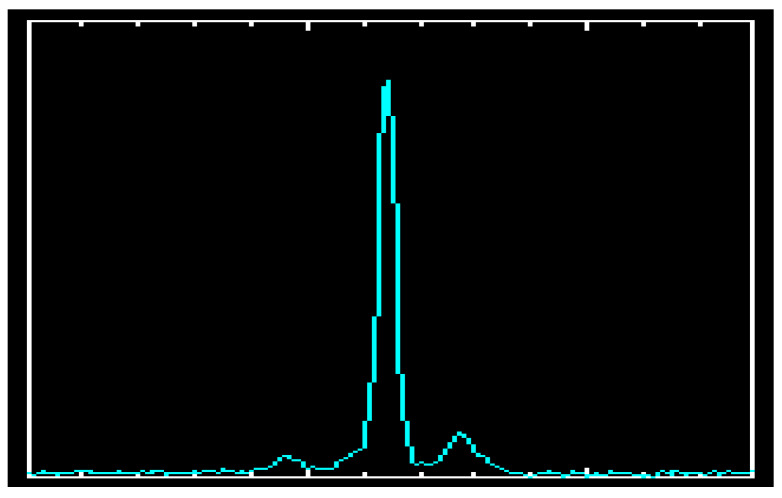
Average pulse of B0329+54.

**Figure 6 sensors-23-07010-f006:**
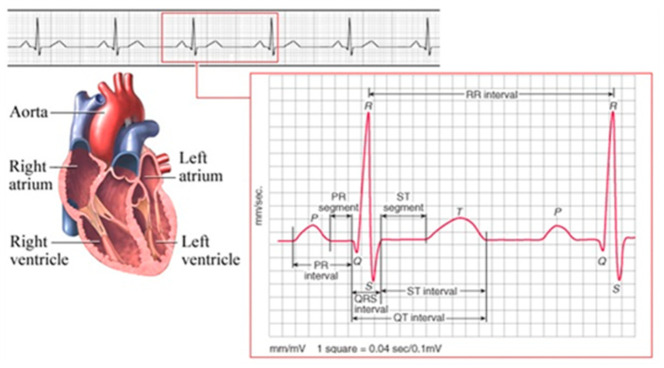
Average pulse of a human heart.

**Figure 7 sensors-23-07010-f007:**
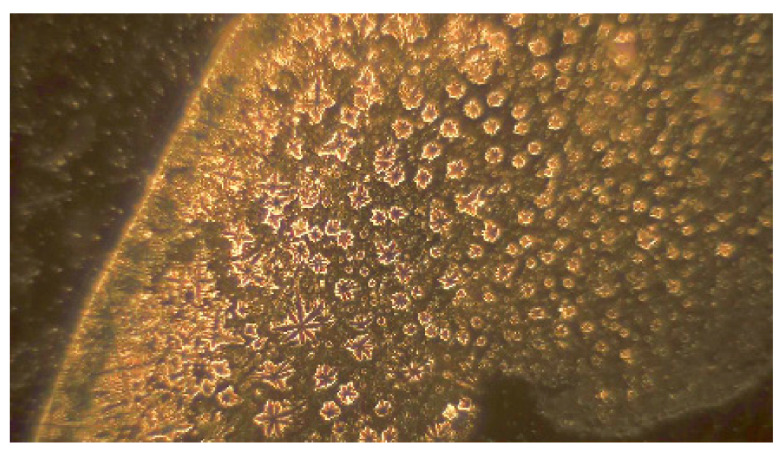
The edge of the drop after irradiation.

**Figure 8 sensors-23-07010-f008:**
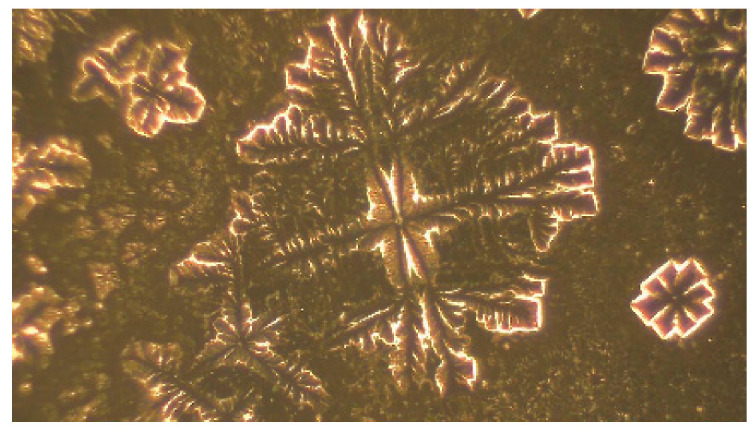
Element of the drop after irradiation.

**Figure 9 sensors-23-07010-f009:**
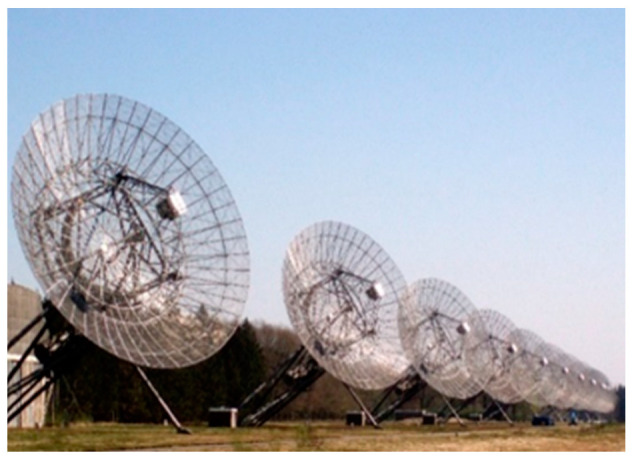
The radio observatory Westerbork.

**Figure 10 sensors-23-07010-f010:**
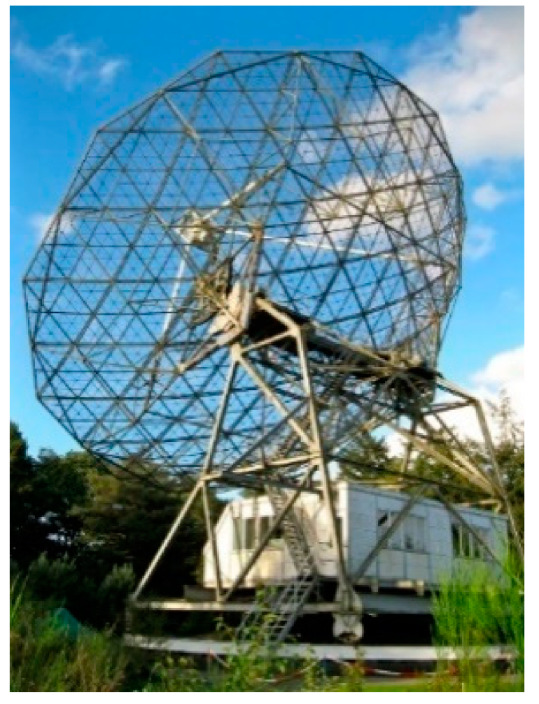
The radio observatory Dwingeloo.

**Figure 11 sensors-23-07010-f011:**
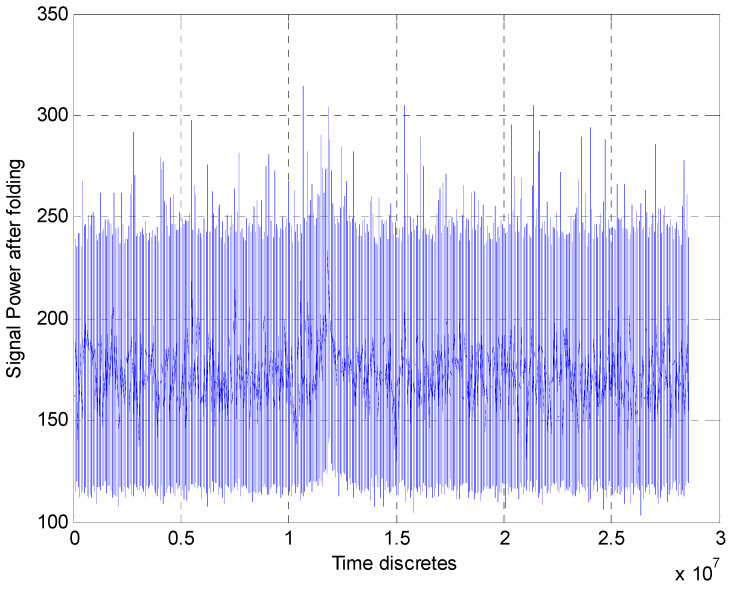
Signal after pulse accumulation (125 periods).

**Figure 12 sensors-23-07010-f012:**
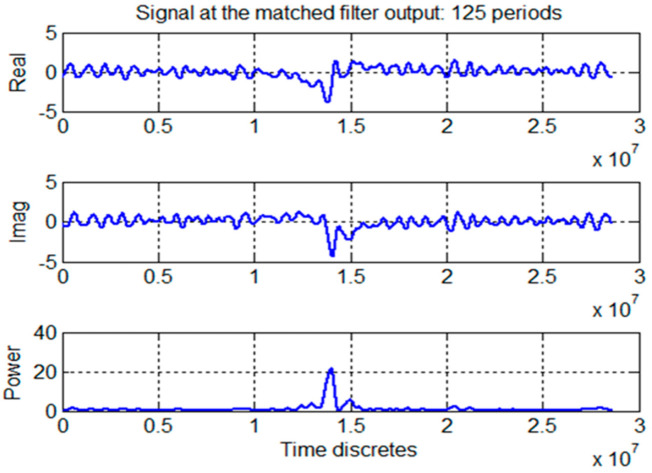
Signals at the input/output of the matched filter (accumulated 125 periods).

**Figure 13 sensors-23-07010-f013:**
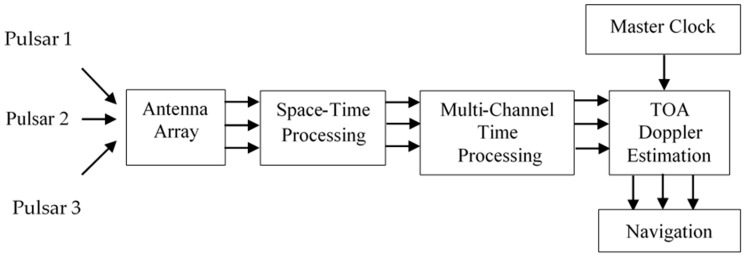
Functional structure of signal processing in a pulsar navigation system.

**Figure 14 sensors-23-07010-f014:**
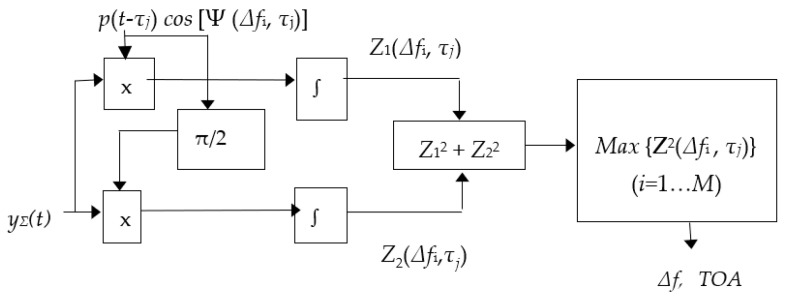
Signal processing in the (*i,j*)-th channel of the correlation estimator of ∆*f* and *τ* in case of the random amplitude and phase.

**Table 1 sensors-23-07010-t001:** Westerbork: pulse accumulation and MF.

N_p_	SNR (dB)	CRLB Estimation		Rough Estimation	
		σ_CRLB_ (τ) (s)	σ_CRLB_(x) (m)	σ(τ) (s)	σ(x) (m)
13	25.2	0.3347 · 10^−5^	1.0041 · 10^3^	1.0821 · 10^−5^	3.2463 · 10^3^
27	27.13	0.2146 · 10^−5^	0.6438 · 10^3^	0.6938 · 10^−5^	2.0814 · 10^3^
41	27.61	0.1922 · 10^−5^	0.5765 · 10^3^	0.6212 · 10^−5^	1.8636 · 10^3^
55	28.38	0.1609 · 10^−5^	0.4828 · 10^3^	0.5203 · 10^−5^	1.5609 · 10^3^
69	28.82	0.1454 · 10^−5^	0.4363 · 10^3^	0.4702 · 10^−5^	1.4106 · 10^3^
83	29.15	0.1348 · 10^−5^	0.4044 · 10^3^	0.4358 · 10^−5^	1.3074 · 10^3^
97	30.07	0.1091 · 10^−5^	0.3272 · 10^3^	0.3526 · 10^−5^	1.0578 · 10^3^
111	29.99	0.1111 · 10^−5^	0.3333 · 10^3^	0.3666 · 10^−5^	1.0998 · 10^3^
125	30.58	0.0970 · 10^−5^	0.2909 · 10^3^	0.3135 · 10^−5^	0.9405 · 10^3^

**Table 2 sensors-23-07010-t002:** Dwingeloo: pulse accumulation and MF.

N_p_	SNR (dB)	CRLB Estimation		Rough Estimation	
		σ_CRLB_ (τ) (s)	σ_CRLB_(x) (m)	σ(τ) (s)	σ(x) (m)
10	14.6	0.3843 · 10^−4^	1.1529 · 10^4^	1.7839 · 10^−4^	5.3515 · 10^4^
50	17.39	0.2021 · 10^−4^	0.6064 · 10^4^	1.2029 · 10^−4^	3.6087 · 10^4^
150	20.26	0.1044 · 10^−4^	0.3132 · 10^4^	0.7935 · 10^−4^	2.3805 · 10^4^

**Table 3 sensors-23-07010-t003:** *Ae* = 10 m^2^: only accumulation.

N_p_	*T_obs_* (min)	SNR (dB)	σ_CRLB_(τ) (s)	σ_CRLB_(x) (m)
83	1	6.70	2.369 · 10^−4^	7.1068 · 10^4^
419	5	10.20	1.059 · 10^−4^	3.1782 · 10^4^
755	9	11.47	0.790 · 10^−4^	2.3689 · 10^4^
1091	13	12.27	0.657 · 10^−4^	1.9711 · 10^4^
1427	17	12.85	0.575 · 10^−4^	1.7236 · 10^4^
1763	21	13.31	0.517 · 10^−4^	1.5508 · 10^4^
2099	25	13.69	0.474 · 10^−4^	1.4214 · 10^4^
2435	29	14.01	0.440 · 10^−4^	1.3197 · 10^4^
2519	30	14.09	0.433 · 10^−4^	1.2975 · 10^4^

## Data Availability

3rd Party Data.

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
