# Peer review of "Investigation of Accuracy of TOA and SNR of Radio Pulsar Signals for Vehicles Navigation"

_sensors, 2023, doi:10.3390/s23157010_

Round 1
Reviewer 1 Report
pulsar B0329+54 should be properly described, including its parameters and observations. The derivation details of most formula are not given, which should be provided. The most references are not related to pulsar navigation detail , which must be improved. The radio frequency for navigation should be given, together with radio equipment.
English is not easily understood.
Author Response
COVER LETTER
with athories’s replies for Reviewer 1
Question: pulsar B0329+54 should be properly described, including its parameters and observations. The radio frequency for navigation should be given, together with radio equipment.
Answer:
The following text is included after Line 406 to 428
When experimentally evaluating the accuracy on the TOA (time delay of the accumulated pulsar signal relative to the pulsar reference signal, i.e. template) are used digital records of the signals obtained in the radio telescopes at the Dwingeloo and Westerbork observatories when observing on the pulsar B0329 + 54, which is a neutron star, emitting the pulse sequence with the e.m. energy spectral density S = 203 mJy at a frequency of 1400 MHz. The profile of the emitted pulses has the following parameters: (i)- repetition period T=0.71452 s, (ii)-pulse width at 10% level T10=31.4 ms; (iii)- pulse width at 50% level T50=6.6 ms. These profile parameters presented in the EPN pulsar database are used in the CRLB estimates of the TOA. To determine the experimental estimates of the TOA, the digital records of pulsar signals obtained by radio telescopes at Dwingeloo and Westerbork observatories are used.
The radio telescope in Dwingeloo uses a single antenna with an area of 500m2 (with gain G[dB]=16.99 dB) while radio telescope Westerbork uses 14 such antennas, which correspond to an antenna with an area of 6000m2 (with gain G[dB]=27.78 dB). From Westerbork, the real data contains the noisy complex baseband signal sampled at frequency of 40 MHz. The frequency band of the input signal is 20MHz with 28582316 samples. The experimental data from Dwingeloo we have fS=70 MHz, the total number of samples of the input data within a single repetition period is N= 499800000.
Signals from pulsar B0329+54 received by radio telescopes before recording pass through the following physical devices: antenna, antenna-feeder device, high and intermediate frequency amplifiers, intermediate frequency filters and ADC). This means that the digital recordings of pulsar signals that are used to estimate the TOA account for all possible signal losses in a real navigation system.
Therefore, the experimental TOA estimates obtained in the paper are realistic estimates because they are based on real pulsar signals in a specific radio telescope. These experimental evaluations realistically reflect the quality of the timing in a radio navigation system using radio telescopes at Dwingeloo and Westerbork observatories.
Question: The derivation details of most formula are not given, which should be provided.
Answer:
In section 4 "Theoretical estimation of the TOA accuracy in a pulsar navigation system", we use Sala's methodology for the potential CRLB estimates. This methodology and resulting formulas were derived by Sala's group and published in [2]. In section 6, we tried to apply a mathematical apparatus for vector description of signals to pulsar signals received at the input of a linear antenna array. The vector description of the received signal allowed us to describe spatio-temporal processing of the pulsar signal in the linear antenna array. First, in each element of the antenna array, the accumulation (folding) of all pulsar pulses received at the input of the antenna array is performed. The next stage of processing is summation of the accumulated signal on all elements of the linear antenna array. First, in each element of the antenna array, the accumulation (folding) of all pulsar pulses received during the observation time at the input of the antenna array is performed. The next stage of signal processing is unification of the signal on all elements of the antenna array. These two stages of processing allow to significantly increase the SNR of the accumulated pulse at the output of the antenna array. The average profile of pulses emitted by a pulsar is known to be one of the main characteristics of every pulsar represented in the EPN database. Using the fact that the average pulse profile from a pulsar is the average of the pulses in the pulse train, we reformulated the signal model at antenna array output as a single radio pulse with two unknown and non-random parameters (TOA and Doppler frequency) and with two random parameters (amplitude and initial phase). This reformulation of the signal model allowed us to solve the task of joint estimation of the signal parameters at the output of the antenna array using the standard maximum likelihood ratio method.
This text was removed from line 587 to 606
According to this algorithm, the sequence of pulsar pulses received in each element of the antenna array can be described by the vector model. This vector model also uses the quasi-deterministic signal model containing unknown and non-random informative parameters, TOA and Doppler frequency, which do not change over the observation time.
More specifically, it is assumed that a sequence of broadband pulses from a pulsar is received by a multi-element antenna array (linear or rectangular). This received signal can be represented as a vector, which allows to mathematically describe it in the form of a product of a complex vector, dependent on time, and a complex vector, not dependent on time, but dependent on the direction of reception of the antenna array. The elements of this vector depend on the phase shift in the different elements of the antenna array, determining the direction of the signal source.
In this paper, we use the quasi-deterministic signal model, with two unknown informative non-random parameters, delay time TOA and Doppler shift of frequency (τ, ∆f) to be estimated, and two random non-informative parameters, amplitude and phase (a , φ0).
The following text is placed instead between Line 587 to 606:
In this chapter, we tried to apply a mathematical apparatus for vector description of signals to pulsar signals received at the input of a linear antenna array. The vector description of the received signal allowed us to describe spatio-temporal processing of the pulsar signal in the linear antenna array. First, in each element of the antenna array, the accumulation (folding) of all pulsar pulses received at the input of the antenna array is performed. The next stage of processing is summation of the accumulated signal on all elements of the linear antenna array. First, in each element of the antenna array, the accumulation (folding) of all pulsar pulses received during the observation time at the input of the antenna array is performed. The next stage of signal processing is unification of the signal on all elements of the antenna array. These two stages of processing allow to significantly increase the SNR of the accumulated pulse at the output of the antenna array. The average profile of pulses emitted by a pulsar is known to be one of the main characteristics of every pulsar represented in the EPN database. Using the fact that the average pulse profile from a pulsar is the average of the pulses in the pulse train, we reformulated the signal model at antenna array output as a single radio pulse with two unknown and non-random parameters (TOA and Doppler frequency) and with two random parameters (amplitude and initial phase). This reformulation of the signal model allowed us to solve the task of joint estimation of the signal parameters at the output of the antenna array using the standard maximum likelihood ratio method.
Question: The most references are not related to pulsar navigation detail, which must be improved.
Answer
Of all the 30 titles listed in in section "Reference", 6 titles comment on the issues of pulsar navigation. Some of the other titles comment on issues of the pulsar timing. The other part of titles is about on the issues of searching and detection of pulsar signals. The last part of the title concerns the issues of statistical processing of radio signals, which we use as a basic theory. As well as a theory of signal processing in antenna arrays. The titles thus presented reflect and support the content of the article. But we have used in our study a large number of other titles concerning pulsar timing and navigation that we have not cited here. For this we add some of them in section 'Reference".
The following text is included after Line 853 to 864.
- Engelen, “Deep Space Navigation System using Radio Pulsars: Front-End”, M. Sc. Thesis, TU Delft, 2009
- KestilΣ, A.A., Engelen, S., Gill, E.K.A.,Verhoeven, C.J.M., Bentum, M.J., Irahhauten, Z.,“An Extensive And Autonomous Deep Space Navigation System Using Radio Pulsars”,Proceedings of the 61st International Astronautical Congress, Prague, 2010, IAC-
- Chaudhri, “Fundamentals, Specifications, Architecture and Hardware towards Navigation System Based on Radio Pulsars”, M.Sc. Thesis, TU Delft, 2011
- Mike Georg Bernhardt, Werner Beckery, Tobias Prinz, Ferdinand Maximilian Breithuth and Ulrich Walter, Autonomous Spacecraft Navigation Based on Pulsar Timing Information, Bernhardt, et. al. 2011, arxiv-version.pdf
- Karunanithi, “A Framework for Designing and Testing the Digital Signal Processing unit of a Pulsar Based Navigation System”, M. Sc. Thesis, TU Delft, 2012
- Lu Wang and Luping Xu, X-Ray Pulsar Signal Detection Based on Time–Frequency Distributions and Shannon Entropy, China Satellite Navigation Conference (CSNC) 2013 Proceedings_ Precise Orbit Determination & Positioning.pdf.

Reviewer 2 Report
The study titled "Investigation of Accuracy of TOA and SNR of Radio Pulsar Signals for Vehicles Navigation" has been reviewed.
Today, the GNSS technique, which has become a standard tool for almost all kinds of positioning studies, is widely and successfully used in almost all kinds of applications. However, the constraints imposed both by the nature of the system and/or by the operator reveal the strategic importance of determining an alternative positioning to this system. This study is very important in this respect. It has the potential to contribute to the literature and at the same time be a trigger for further studies on this subject.
It would be appropriate to make the following some items/corrections:
1-) Statement of problem, research gap and objectives of the study should be stated more clearly.
2-) In the study, the weaknesses of GNSS can be specified and it can be mentioned which weaknesses of GNSS can be eliminated by the handled method.
3-) It would be useful to estimate more realistic accuracy (not precision). It would be appropriate to provide more satisfactory information on this subject.
4-) Figure 3 is not readable.
5-) The content in Figure 6 should be in English (probably Cyrillic),
6-) Page 5, CNSS or GNSS on Line 207?
7-) Page 6, Line 226, parenthesis not closed.
8-) In[?] on Page 7, Line 289 should be corrected,
9-) Instead of "In this paragraph" in the study, "In this chapter" may be more appropriate.
Author Response
COVER LETTER
with athories’s replies for Reviewer 2
Question-1: Statement of problem, research gap and objectives of the study should be stated more clearly.
Answer-1:
The following text in inserted after Line 64.
In the proposed article, the theory and principles of pulsar space navigation proposed in works [1,2] are used. We take into account that the navigation pulsar system offered by us will not work in the cosmic space, but on the earth's surface or in the air. That made us search for similarities with the principles and theory of GNSS radio navigation that is so widely used at present. According to the concept of the structure of a pulsar navigation system proposed in works [1,2], it consists of 4 parts: instrumental stage: antennas; timing estimation stage: estimation of pulsar timing parameters; position estimation stage: determination of position; navigation stage: higher levels of navigational information. Our scientific task in this article is in the field of receiving and signal processing of pulsar signals for the needs of transport navigation, (antennas; timing estimation stage,).
In our research, we use the well-known rougher temporal (not phased) approach to transport navigation that was successfully used in early versions of GNSS, because unlike artificial signals for positioning in GNSS, pulsar signals are not modulated and it is impossible to identify the number of phase cycles (pulsar periods) between two signal positions.
Our previous research results are also commented on in the present article in chapter 3. They have been connected with the development of signal algorithms including: folding in the time domain, matched filtering of the pulsar signal to extract the signal from the noise and cross-correlation of the pulsar signal with the pulsar “template” (for estimation of TOA) for processing the recordings from two radio telescopes of the studied pulsar signal B0329+54. A rough estimate of TOA measurement accuracy, or pulsar timing accuracy, we have estimated according to the approach proposed by Loriner [1] with the width of the cross-correlation peak at the 0.9 level. Numerical results were obtained in these previous papers on the relationship between the number of accumulated pulses at the folding and the SNR, and the accuracy of the TOA estimation as a function of the SNR, for both types of pulsar signal recordings from a radio telescope with a very large antenna around 6000 sq. m (Westerbork), and a much smaller antenna about 500 sq.m (Dwingeloo). From our results and the results obtained in the project, it followed that the accuracy of the TOA estimation by time / distance is extremely insufficient for real-time navigation of an aircraft. These studies showed that the proposed concept of a pulsar navigation system with a small antenna working in real time for the needs of aircraft (transport) navigation, using a mirror antenna and standard pulsar timing, is not adequate. Because the energy from a pulsar received within a few minutes, using a small antenna, is insufficient to perform a quality pulsar timing, guaranteeing the required accuracy of the TOA measurement.
This challenge led us in the present article to think about what known statistical methods and signal processing approaches can be used, to maximize as much as possible the receiving pulsar energy at the input of the small antenna device, and to speed up and improve the quality of pulsar timing, for a few minutes. Since we work in the field of radar signal processing, which is designed to work in real time, we propose to use its concepts, approaches and algorithms. Most of the known radar signal processing tasks, such as detection, signal parameter estimation, velocity distance and phase estimation, in the statistical theory are solved and developed for different models of signal and interference.
We first checked in the present paper (in chapters 4 and 5) the quality of the signal algorithms proposed and developed by us for assessing the accuracy of TOA estimation presented in previous publications. The verification consisted of evaluating their proximity to the potential CLBR accuracy estimates, at the same SNR ratios as for the recorded pulsar signals. In Chapter 4, a potential TOA estimate was proposed to be calculated as a Crammer-Law Low Bound (CRLB-estimate), the calculation of which also requires an estimate of the SNR level and the parameters of the profile of the pulsar B0329+54, taken from the European pulsar database. From the results obtained in chapter 5, it follows that the rough estimates of TOA obtained by us, at high levels of SNR are close to the potential CRLB- estimates. In order to maximize as much as possible, the received pulsar energy at the input of a small antenna device, we propose in the present paper (Chapter 6) to the used radio astronomy pulsar timing approach, including antenna, receiver gain, dedispersion, and folding, to add all possible radar approaches for receiving signals together with their algorithms, adapted to the specifics of pulsar signals to be used in the processing in the antenna system.
The most used and well-known radar mathematical approach for the description of signal processing in the antenna arrays is the vector representation of e/m field of the pulsar signal for the needs of the spatial processing in the linear antenna arrays. This presentation of e/m pulsar single signal field describes it with a very large number of points, in contrast to the single point description of the pulsar signal field used in mirror antennas. It allows more effective optimal spatial (with antenna arrays) processing of the pulsar signals, and increases the SNR of the received spatial energy. Following this approach, we propose in this paper the simultaneous use of several antenna arrays, matched filtering with the shape of the pulsar signal in each channel of the linear antenna array and multi-frequency simultaneous reception of the pulsar signals in these antenna arrays from the same pulsar. All these signal processing jointly increase the maximum possible SNR ratio at the output of the antenna system. In order to satisfy the requirements for real-time operation in the estimation of the parameters of pulsar signals (TOA and Doppler shift), we propose to use a multi-channel (parallel) approach. It allows at the same time to perform the estimation of the Doppler velocity of the vehicle in different frequency channels of the measurement frequency interval. In each of these frequency channels, TOA estimation is performed simultaneously in time channels of the measured time interval.
The proposed ground navigation algorithm (including antenna and pulsar timing) is developed under the following constraints. Appropriate measures have been taken for stabilization of the received phase pulsar pulses in the navigation receiver, as is done in radars (systems of phase autoadjustment), to measure the speed of a moving vehicle. The algorithm for dedispersion of pulsar signals is assumed to be available in the spatial-time processing channels and is not considered separately in the proposed article.
Question-2: In the study, the weaknesses of GNSS can be specified and it can be mentioned which weaknesses of GNSS can be eliminated by the handled method.
Answer-2:
The following text is included after Line 195
The development of applications in GSM from the 5th generation (smart transport, cars, city) allow to use the information received from the GNSS navigation systems even more precisely. Together with other sensors (lidars, radars, systems embedded in vehicles and on roads), systems for street traffic and urban transport perform the control of moving vehicles without a pilot.
That is, the complexing of many sensors and sensors for information, in systems for positioning, management and navigation is carried out. Similar systems were built initially in maritime traffic, then in aviation, and now it is realized in land transport. A disadvantage of this complex multi-sensor monitoring and control system consisting of navigation networks (space and ground) is its high sensitivity to radio interference created unintentionally or intentionally, especially in densely populated areas. However, the major drawback of this high-precision radio navigation network is that it does not have a wide coverage of the earth's surface, and especially of the sea. Its 3, 4, and 5 generation GSM terrestrial part is primarily located in densely populated areas and large cities.
Therefore, it is very important to look for other alternative systems for passive navigation covering the entire globe, and which use other more suitable radio signals, for example, of space origin.
In this article, a suitable source of space radio signals - pulsars - is chosen for terrestrial navigation. It is currently the only source of radio signals in space used for spacecraft navigation.
Like any terrestrial radionavigation system, this system using pulsar radio signals will be sensitive to radio interference. However, it will be independent of any human intervention (in case of failure or shutdown of the GNSS systems). Incorporating such a coarser pulsar navigation system into the currently existing high-precision multi-sensor navigation system for monitoring and control could be very important.
Question-3: It would be useful to estimate more realistic accuracy (not precision). It would be appropriate to provide more satisfactory information on this subject.
Answer-3:
The following text is included after Line 331.
When experimentally evaluating the accuracy on the TOA (time delay of the accumulated pulsar signal relative to the pulsar reference signal, i.e. template) are used digital records of the signals obtained in the radio telescopes at the Dwingeloo and Westerbork observatories when observing on the pulsar B0329 + 54, which is a neutron star, emitting the pulse sequence with the e.m. energy spectral density S = 203 mJy at a frequency of 1400 MHz. The profile of the emitted pulses has the following parameters: (i)- repetition period T=0.71452 s, (ii)-pulse width at 10% level T10=31.4 ms; (iii)- pulse width at 50% level T50=6.6 ms. These profile parameters presented in the EPN pulsar database are used in the CRLB estimates of the TOA. To determine the experimental estimates of the TOA, the digital records of pulsar signals obtained by radio telescopes at Dwingeloo and Westerbork observatories are used.
The radio telescope in Dwingeloo uses a single antenna with an area of 500m2 (with gain G[dB]=16.99 dB) while radio telescope Westerbork uses 14 such antennas, which correspond to an antenna with an area of 6000m2 (with gain G[dB]=27.78 dB).
Signals from pulsar B0329+54 received by radio telescopes before recording pass through the following physical devices: antenna, antenna-feeder device, high and intermediate frequency amplifiers, intermediate frequency filters and ADC). This means that the digital recordings of pulsar signals that are used to estimate the TOA account for all possible signal losses in a real navigation system.
Therefore, the experimental TOA estimates obtained in the paper are realistic estimates because they are based onreal pulsar signals in a specific radio telescope. These experimental evaluations realistically reflect the quality of the timing in a radio navigation system using radio telescopes at Dwingeloo and Westerbork observatories.
Question-4: Figure 3 is not readable.
Answer-4: Figure 3 is enlarged and placed on a separate line
Question-5: The content in Figure 6 should be in English (probably Cyrillic),
Answer-5: Figure 6 is replaced by a readable one.
Question-6: CNSS or GNSS on Line 207?
Answer-6: In Line 207, the text “CNSS” is replaced by “GNSS”
Question-7: Line 226, parenthesis not closed.
Answer-7: In Line 226, the text “(up to 10 sq. m|” is replaced by “(up to 10 sq. m)”
Question-8: In[?] on Page 7, Line 289 should be corrected,
Answer-8: In Line 289, the text “[?]” is replaced by “[14,15]”
Question-9: Instead of "In this paragraph" in the study, "In this chapter" may be more appropriate.
Answer-9: Yes. Throughout the article, the text “in this paragraph” is replaced by “in this chapter”

Reviewer 3 Report
This is an interesting topis with plenty of room for improvements. There are several comments regarding this paper.
· In Abstract, Line 16, the “received by pulsars” should be replaced by “received from pulsars”.
· The list of keywords should be extended with the word “navigation”.
· In Introduction, line 46, abbreviation CNSS is used, without clarification (maybe should have been GNSS). So are abbreviations for Cramér–Rao lower bound etc. Write in full words all abbreviation on first appearance.
· In Section 3 the paragraphs are non-coherently following one another and lacking the thoughts flow. End of Section 2 and Section 3 should be more focused on the paper topic.
· In Line 289 the literature reference is broken.
· It is not clear which pulsar signal strength is sufficient to be received with this algorithm (good-enough SNR) using the 10 square meters antenna, and how many pulsars meet such criteria.
· What is the required unobstructed reception time and does it truly make this algorithm applicable for real-time navigation?
· At the end of Section 6 it would be good to see a practical example of algorithm with real-world numbers to back up its applicability.
· Literature [12] is missing a title.
· In Figure 15 there is a spelling error in the second rectangle. The Fig. 15 is also very crude, with errors misalignment and the caption missing a blank space.
· The rest of the paper has dozens of double blank spaces.
Author Response
COVER LETTER
with athories’s replies for Reviewer 3.New
Dear Reviewer 3.New, I am sending to you the revised version of the paper , which includes all the answers to the rest of the reviewers 1, 2 and 4, as well as the answers to you.
Question: In Abstract, Line 16, the “received by pulsars” should be replaced by “received from pulsars”.
Answer: In Abstract, Line 16, the “received by pulsars” is replaced by “received from pulsars”.
Question: The list of keywords should be extended with the word “navigation”.
Answer: The list of keywords is extended with the word “navigation”.
Question: In Introduction, line 46, abbreviation CNSS is used, without clarification (maybe should have been GNSS). So are abbreviations for Cramér–Rao lower bound etc. Write in full words all abbreviation on first appearance.
Answer: In Line 46, “CNSS” is replaced by “GNSS”.
On first appearance the abbreviations SNR, TOA and CRLB are written in full words.
Question: · In Section 3 the paragraphs are non-coherently following one another and lacking the thoughts flow. End of Section 2 and Section 3 should be more focused on the paper topic.
Answer:
The text from line 406 to 416 with the answers to the previous reviewers 1, 2 and 4 is changed and moved
to line 190 to 195 including an answer to you as well.
The pulsar B0329 + 54, which is a neutron star, emitting the pulse sequence with the e.m. energy spectral density S = 203 mJy at a frequency of 1400 MHz. The profile of the emitted pulses has the following parameters: (i)- repetition period T=0.71452 s, (ii)-pulse width at 10% level T10=31.4 ms; (iii)- pulse width at 50% level T50=6.6 ms. These profile parameters presented in the EPN pulsar database are used in the CRLB estimates of the TOA.
The text from line 225 to 228 with the answers to the previous reviewers 1, 2 and 4 is moved to line 211 to 217 including an answer to you as well.
The following text is not related to navigation. However, it expands the known facts about pulsar B0329+54, discovers new facts that may be useful to the wider scientific community in the field of signal processing, expanding the knowledge of the properties of pulsar signals. The obvious connection presented in fig. 5 and fig.6 between the pulses of the human heart and the researched by us pulsar B0329+54, and a microscopic photo fig. 7 and fig.8 of a water drop irradiated with sound from pulsar B0329+54s the result of our research on the properties of pulsar signals in the last 5 years.
And to improve the logic in the paper, we moved the text from line 406 to 416 with the answers to the previous reviewers 1, 2 and 4 to line 250 to 259.
In our previus paper [15, 17-21], are used digital records of the signals obtained in the radio telescopes at the Dwingeloo and Westerbork observatories, who observing on the pulsar B0329 + 54, for experimentally evaluating the accuracy on the TOA (time delay of the accumulated pulsar signal relative to the pulsar reference signal, i.e. template).
The pulsar B0329 + 54, which is a neutron star, emitting the pulse sequence with the e.m. energy spectral density S = 203 mJy at a frequency of 1400 MHz. The profile of the emitted pulses has the following parameters: (i)- repetition period T=0.71452 s, (ii)-pulse width at 10% level T10=31.4 ms; (iii)- pulse width at 50% level T50=6.6 ms. These profile parameters presented in the EPN pulsar database are used in the CRLB estimates of the TOA.
And to improve the logic of the paper, the description of the radiotelescopes is moved right below their photos. Therefore, the text from line 417 to 423 with the answers to the previous reviewers 1, 2 and 4 is moved to line 332 to 340.
The radio telescope in Dwingeloo uses a single antenna with an area of 500m2 (with gain G[dB]=16.99 dB) while radio telescope Westerbork uses 14 such antennas, which correspond to an antenna with an area of 6000m2 (with gain G[dB]=27.78 dB). The real data from Westerbork contains the noisy complex baseband signal sampled at frequency of 40 MHz. The frequency band of the input signal is 20MHz with 28582316 samples. The experimental data from Dwingeloo is sampled at frequency of fS=70 MHz, the total number of samples of the input data within a single repetition period is N= 499800000.
Question: In Line 289 the literature reference is broken.
Answer:
In Line 365,”In[?]’ is replaced by “In[20,21]”
Question: It is not clear which pulsar signal strength is sufficient to be received with this algorithm (good-enough SNR) using the 10 square meters antenna, and how many pulsars meet such criteria.
Answer:
In the paper is written in the Line 94 -101;
From our results and the results obtained in the project, it followed that the accuracy of the TOA estimation by time / distance is extremely insufficient for real-time navigation of an aircraft. These studies showed that the proposed concept of a pulsar navigation system with a small antenna (about tens to a hundred square meter antenna) working in real time for the needs of aircraft (transport) navigation, using a mirror antenna and standard pulsar timing, is not adequate. Because the energy from a pulsar received within a few minutes, using a small antenna, is insufficient to perform a quality pulsar timing, guaranteeing the required accuracy of the TOA measurement.
In the paper is written in the Line 320-327;
“In [20, 21], when calculating the theoretical CRLB estimates of the TOA, a methodology of Sala was used, which takes into account the parameters of the signals from three pulsars (from the EPN pulsar database), the receiver parameters to determine the output signal-to-noise ratio (antenna of 10 sq.m, system temperature T= 15-150 K), and also signal processing with accumulation [17]. The results obtained in [20, 21] show that the accuracy of distance determination in the aviation navigation system using signals from 3 pulsars is of the order of tens of kilometers. This is grossly inadequate for aircraft navigation, and is perhaps more suitable for marine navigation.
In the paper is written in the Line 516-534;
Table 3 presents CRLB estimates of TOA accuracy calculated by the Sala's method according to expression (1) depending on the SNR values, the corresponding observation time of the pulsar B0329+54, for the case of a small antenna (10 sq.m ) suitable for the use in a navigation system of transport vehicles.
The CRLB estimates of TOA presented in Table 3 show that using the small antenna (10 sq.m ) and operating in the real-time mode (1-5 min) the error in distance (from TOA) is of about 31782 m. This is extremely insufficient for the needs of the transport navigation system. The experimental CRLB estimates of the TOA accuracy presented in this paper are similar to the theoretical CRLB estimates given in [20,21].
This accordance leads to the same conclusions made in [17,20,21] that the accuracy of the TOA estimate for pulsar B0329+54 is grossly inadequate for navigation of aerplanes using the classical TOA measurement approach, with a low-noise radio telescope receiver and a mirrored small antenna. These conclusions are analogous to the conclusions in [2] that such a radio navigation system with a mirror antenna of 10 sq.m is highly ineffective for space navigation by pulsars. It is necessary to look for other possible ways to increase the received energy from pulsars.
About :(how many pulsars meet such criteria) In the paper is written in the Line 170 -187;
The detailed analysis of most known pulsars, aiming to determine the most suitable of them for space navigation, was carried out in [2]. The analysis of pulsars is made according to several criteria concerning the energy of the emitted signals, the frequency of following pulses, the shape of the average pulse, etc. As a result of analysis, 50 pulsars have been identified, the signals of which are most suitable for space navigation. This number of pulsars is comparable to the total number of satellites used in modern GNSS (GPS, GLONAS, BeiDOU) [5]. The distribution of 50 pulsars in space elliptical and galactic coordinates, whose signals are most suitable for navigation is schematically illustrated in Figure 1 and Figure 2 [2].
180 150 120 90 60 30 0 330 300 270 240 210 180 360 330 300 270 240 210 180 150 120 90 60 30 0
Galactic longitude Ecliptic longitude
Figure. 1 Positions in galactic coordinates Figure 2. Ecliptic coordinates
As can be seen from Figure 1 and Figure 2, the distribution of pulsars covers almost the entire space and, therefore, pulsars such as GNSS satellites can be used for space navigation.
Question: What is the required unobstructed reception time and does it truly make this algorithm applicable for real-time navigation?
Answer:
In the paper is written in the Line 94 -107;
From our results and the results obtained in the project, it followed that the accuracy of the TOA estimation by time / distance is extremely insufficient for real-time navigation of an aircraft. These studies showed that the proposed concept of a pulsar navigation system with a small antenna (about tens to a several hundred square meter) working in real time for the needs of aircraft (transport) navigation, using a mirror antenna and standard pulsar timing, is not adequate. Because the energy from a pulsar received within a few minutes, using a small antenna, is insufficient to perform a quality pulsar timing, guaranteeing the required accuracy of the TOA measurement.
This challenge led us in the present article to think about what known statistical methods and signal processing approaches can be used, to maximize as much as possible the receiving pulsar energy at the input of the small antenna device, and to speed up and improve the quality of pulsar timing, for a few minutes. Since we work in the field of radar signal processing, which is designed to work in real time, we propose to use its concepts, approaches and algorithms.
In the paper is written in the Line 577 -582;
In the transport navigation system operating in the real-time mode, the use of the classical approach in radio telescopes, which requires the accumulation (folding) of the energy of pulsar signals over hours, is very unacceptable. This requirement for real-time operation necessitates the search for other known statistical approaches for the synthesis of a suitable complex real-time pulsar signal processing algorithm that allows obtaining a rough-accuracy estimate of TOA for ground transportation navigation needs.
In the paper is written in the Line 591 -599;
In this chapter, a radar approach is proposed for the synthesis of optimal processing in antenna arrays of a series of vector signals from pulsars for the needs of transport navigation in the real-time mode. Using the approach of vector time functions in the synthesis of optimal signal processing systems allows to simplify the complex theory of optimal processing of random electromagnetic fields. This field-to-vector transformation agrees well with the real electromagnetic field discretization performed in the multi-element antennas (antenna arrays). As already mentioned, the solving the most difficult problem for transport navigation with pulsars is in the search for new ways to quickly accumulate the necessary SNR of the received pulsar signals [17,20,21].
Question: At the end of Section 6 it would be good to see a practical example of algorithm with real-world numbers to back up its applicability.
Answer:
In space vehicles, the size of the antenna is limited and therefore the observation time window when working in the real-time mode is larger to realize the necessary positioning accuracy. The picture below shows that the accuracy of the space navigation systems compared to navigation systems on earth is closer to these systems: INS (Typical Inertial Navigation Systems) and Omega/VLF and Loran, several nautical miles (1 to 10).
The following text is included after Line 536 to 572;
It is necessary to look for other possible ways for real-time pulsar navigation. In the last 10-15 years, the positioning and navigation in space of spacecrafts is done only with the use of signals from X-ray pulsars. These are real-time systems. The required accuracy of space positioning and navigation is of the order of tens or several kilometers depending on the size of the spacecraft and its antenna and system. Accordingly, the real-time operation requirements are different compared to those for terrestrial navigation systems. In space navigation systems, this time window defining the real-time operation is orders of magnitude larger (tens of minutes and hours). The spatial accuracy for positioning is many times smaller than that of the Earth's surface, which is currently used in GNSS (tens of meters, and meters). I.e. the difference in positioning accuracy is several (3-4) orders of magnitude.
As shown in the proposed paper, we have calculated a rough estimate of the TOA accuracy according to Loriner's approach given in [1] as the ratio of the width of the autocorrelation peak at the 0.7 level to the SNR. The X-ray pulsar autocorrelation peak is orders of magnitude narrower at small SNR ratios than that of the B0329+54 pulsar we used. Therefore, positioning accuracy in space using X-rays (not available on Earth) is higher than on Earth using radio pulsars.
The magnitude of the SNR depends on the size of the antenna, the specifics of signal processing in the antenna, the capabilities of the receiving device and the observation time. Because the capabilities of the receiving device and the processing in the antenna are usually known in advance, the increase in SNR is possible to obtain only from the size of the antenna and the observation time. Therefore, if we want to offer a pulsar navigation system with a higher accuracy of measuring the position of the vehicle, to ensure its operation, a longer monitoring time of the pulsar signal is needed to accumulate the required SNR level.
The observation time window required to achieve a high level of SNR can be substantially large (minutes, tens of minutes, or hours). When working in the real-time mode, depending on the type of vehicle, a compromise must be found between the positioning accuracy and the requirement of real-time operation of a specific vehicle (ship, cars, etc.).
In space vehicles, the size of the antenna is limited and therefore the observation time window when working in the real-time mode is larger to realize the necessary positioning accuracy. The positioning accuracy of space navigation pulsar systems compared to navigation systems on earth is closer to these systems: INS (Typical Inertial Navigation Systems) and Omega/VLF and Loran.
Therefore, from everything shown above, it is not possible for the accuracy of the position from the land navigation system for the transport navigation to be better than the accuracy from the space navigation system.
Question: · Literature [12] is missing a title.
Answer: In Reference, the omitted title is inserted.
Question: In Figure 15 there is a spelling error in the second rectangle. The Fig. 15 is also very crude, with errors misalignment and the caption missing a blank space.
Answer: The Figure 13 is corrected.

Reviewer 4 Report
Sorry, but our text did not looks like as serious scientific work. You understand physics (and mathematics) of pulsars very superficially. Text within rows 150-194 is absolutely inappropriate for paper about "navigation". It must be exluded!
Figures 1 and 2 are wrong. What does it mean "elliptic longitude"? Ecliptic or equatorial? Fig. 1 seems to be just in Galactic coordinates. TOA - is not a delay - it is time of arrival of a single pulse! And so on... You must have !accurate ephemerides! to find a delay. And you must recalculate the TOA in baricentric coordinates. That is not trivial task for any navagation systems.
Use the professional astronomical language for description of pulsar properties.
Author Response
COVER LETTER
with athories’s replies for Reviewer 3
Question
Sorry, but our text did not look like as serious scientific work. You understand physics (and mathematics) of pulsars very superficially.
Answer:
Dear Reviewer 3, you are absolutely right about the physics and math of pulsars.
We are not specialists in the physics and mathematics of pulsars, because we are not astrophysicists. Part of our team has been working in the scientific field of radar signal processing for over 30 years, as well as we have participated in military industry developments.
In 2013 we participated in the European project, "Pulsar Plane", with the signal processing of the recorded pulsar signals at the Westerbork and Dwingeloo radio telescopes in the Netherlands. The purpose was to estimate the accuracy of TOA measurement for the needs of aircraft navigation. The results of our research were successfully published at a number of European conferences.
Question
Text within rows 150-194 is absolutely inappropriate for paper about "navigation". It must be excluded!
Answer:
You are right, this text does not address navigation issues. This text examines various little-known aspects of pulsar signal processing. For this purpose, we have proposed it there (lines 150-194). We will exclude the lines concerning Prof. Knepitz's research, regarding the influence of e.m.v. on the water drop.
The obvious connection presented in fig. 5 and fig.6 between the pulses of the human heart and the researched by us pulsar B0329+54, is the result of our research on the properties of pulsar signals in the last 5 years, in different scientific projects. Fig. 7 and Fig. 8, a microscopic photo of a water drop irradiated with sound from pulsar B0329+54, made by the researcher I. Todorov, is presented. This is also part of our latest research.
The text in Lines 161-175, concerning Prof. Knepitz's research on the influence of the electromagnetic waves of the water drop is removed.
The text in Lines 225-228 is corrected like this:
The following text is not related to navigation. However, it expands the known facts about pulsar B0329+54, discovers new facts that may be useful to the wider scientific community in the field of signal processing, expanding the knowledge of the properties of pulsar signals.”
Question
Figures 1 and 2 are wrong. What does it mean "elliptic longitude"? Fig. 1 seems to be just in Galactic coordinates.
Answer:
Yes, you are right, the places of the inscriptions have been changed in relation to figures 1 and 2.
The text at row 181 , 182 is corrected as:
Galactic longitude Ecliptic longitude,
Figure. 1 Positions in galactic coordinates Figure 2. Positions in ecliptic coordinates
Question:
What does it mean "elliptic longitude"? Ecliptic or equatorial?
Answer:
We were wrong, it is about Ecliptic longitude
The text at row 181 , 182 is corrected as:
Ecliptic longitude
Question
TOA - is not a delay - it is time of arrival of a single pulse! And so on..
You must have !accurate ephemerides! to find a delay. And you must recalculate the TOA in baricentric coordinates. That is not trivial task for any navigation systems.
Answer:
In the proposed article, (as well as the European project of 2013), the theory and principles of pulsar space navigation proposed in works [1,2] are used, 1. Loriner, D.; Kramer, M. Handbook of pulsar astronomy, Cambridge university press, N. Y., 2005., and [2], “Sala, J et al. Feasibility study for a spacecraft navigation system relying on pulsar timing information. In Tech. Rep. 03/4202, ARIADNA Study, June 2004.
Of course, we take into account that the navigation pulsar system offered by us will not work in the cosmic space, but on the earth's surface or in the air. That made us search for similarities with the principles and theory of GNSS radio navigation that is so widely used at present.
According to the concept of the structure of a pulsar navigation system proposed in works [1,2], it consists of 4 parts: instrumental stage: antennas; timing estimation stage: estimation of pulsar timing parameters; position estimation stage: determination of position; navigation stage: higher levels of navigational information.
Unfortunately, we may not have outlined well the exact area of our investigation in terms of pulsar navigation. Our scientific task in this article is in the field of receiving and signal processing of pulsar signals for the needs of transport navigation, (antennas; timing estimation stage,).
In our research, we use the well-known rougher temporal (not phased) approach to transport navigation that was successfully used in early versions of GNSS, because unlike artificial signals for positioning in GNSS, pulsar signals are not modulated and it is impossible to identify the number of phase cycles (pulsar periods) between two signal positions.
Our previous research results (in the PulsarPlane project, published in a number of articles) are also commented on in the present article in chapter 3. They have been connected with the development of signal algorithms including: folding in the time domain, matched filtering of the pulsar signal to extract the signal from the noise and cross-correlation of the pulsar signal with the pulsar “template” (for estimation of TOA) for processing the recordings from two radio telescopes of the studied pulsar signal B0329+54. A rough estimate of TOA measurement accuracy, or pulsar timing accuracy, we have estimated according to the approach proposed by Loriner with the width of the cross-correlation peak at the 0.9 level. According to Loriner, "TOA can be calculated accurately by cross-correlating the observed pulsar profile with a high signal-to-noise (S/N) 'template' profile obtained by adding many earlier observations at the particular observing frequency."
Numerical results were obtained in these previous papers on the relationship between the number of accumulated pulses at the folding and the SNR, and the accuracy of the TOA estimation as a function of the SNR, for both types of pulsar signal recordings from a radio telescope with a very large antenna around 6000 sq. m (Westerbork), and a much smaller antenna about 500 sq.m (Dwingeloo). From our results and the results obtained in the project, it followed that the accuracy of the TOA estimation by time / distance is extremely insufficient for real-time navigation of an aircraft.
These studies showed that the proposed concept of a pulsar navigation system with a small antenna working in real time for the needs of aircraft (transport) navigation, using a mirror antenna and standard pulsar timing, is not adequate. Because the energy from a pulsar received within a few minutes, using a small antenna, is insufficient to perform a quality pulsar timing, guaranteeing the required accuracy of the TOA measurement.
This challenge led us in the present article to think about what known statistical methods and signal processing approaches can be used, to maximize as much as possible the receiving pulsar energy at the input of the small antenna device, and to speed up and improve the quality of pulsar timing, for a few minutes. Since we work in the field of radar signal processing, which is designed to work in real time, we propose to use its concepts, approaches and algorithms.
Most of the known radar signal processing tasks, such as detection, signal parameter estimation, velocity distance and phase estimation, in the statistical theory are solved and developed for different models of signal and interference.
We first checked in the present paper (in chapters 4 and 5) the quality of the signal algorithms proposed and developed by us for assessing the accuracy of TOA estimation presented in previous publications. The verification consisted of evaluating their proximity to the potential CLBR accuracy estimates, at the same SNR ratios as for the recorded pulsar signals.
In Chapter 4, a potential TOA estimate was proposed to be calculated as a Crammer-Law Low Bound (CRLB-estimate), the calculation of which also requires an estimate of the SNR level and the parameters of the profile of the pulsar B0329+54, taken from the European pulsar database. From the results obtained in chapter 5, it follows that the rough estimates of TOA obtained by us, at high levels of SNR are close to the potential CRLB- estimates.
In order to maximize as much as possible, the received pulsar energy at the input of a small antenna device, we propose in the present paper (Chapter 6) to the used radio astronomy pulsar timing approach, including antenna, receiver gain, dedispersion, and folding, to add all possible radar approaches for receiving signals together with their algorithms, adapted to the specifics of pulsar signals to be used in the processing in the antenna system.
The most used and well-known radar mathematical approach for the description of signal processing in the antenna arrays is the vector representation of e/m field of the pulsar signal for the needs of the spatial processing in the linear antenna arrays. This presentation of e/m pulsar single signal field describes it with a very large number of points, in contrast to the single point description of the pulsar signal field used in mirror antennas. It allows more effective optimal spatial (with antenna arrays) processing of the pulsar signals, and increases the SNR of the received spatial energy.
Following this approach, we propose in this paper the simultaneous use of several antenna arrays, matched filtering with the shape of the pulsar signal in each channel of the linear antenna array and multi-frequency simultaneous reception of the pulsar signals in these antenna arrays from the same pulsar. All these signal processing jointly increase the maximum possible SNR ratio at the output of the antenna system.
In order to satisfy the requirements for real-time operation in the estimation of the parameters of pulsar signals (TOA and Doppler shift), we propose to use a multi-channel (parallel) approach. It allows at the same time to perform the estimation of the Doppler velocity of the vehicle in different frequency channels of the measurement frequency interval. In each of these frequency channels, TOA estimation is performed simultaneously in time channels of the measured time interval. The proposed ground navigation algorithm (including antenna and pulsar timing) is developed under the following constraints. Appropriate measures have been taken for stabilization of the received phase pulsar pulses in the navigation receiver, as is done in radars (systems of phase autoadjustment), to measure the speed of a moving vehicle. The algorithm for dedispersion of pulsar signals is assumed to be available in the spatial-time processing channels and is not considered separately in the proposed article.
The text at row 65-90 is removed. Instead this new text is placed at row 65-146:
In the proposed article, the theory and principles of pulsar space navigation proposed in works [1,2] are used. We take into account that the navigation pulsar system offered by us will not work in the cosmic space, but on the earth's surface or in the air. That made us search for similarities with the principles and theory of GNSS radio navigation that is so widely used at present. According to the concept of the structure of a pulsar navigation system proposed in works [1,2], it consists of 4 parts: instrumental stage: antennas; timing estimation stage: estimation of pulsar timing parameters; position estimation stage: determination of position; navigation stage: higher levels of navigational information. Our scientific task in this article is in the field of receiving and signal processing of pulsar signals for the needs of transport navigation, (antennas; timing estimation stage,).
In our research, we use the well-known rougher temporal (not phased) approach to transport navigation that was successfully used in early versions of GNSS, because unlike artificial signals for positioning in GNSS, pulsar signals are not modulated and it is impossible to identify the number of phase cycles (pulsar periods) between two signal positions.
Our previous research results are also commented on in the present article in chapter 3. They have been connected with the development of signal algorithms including: folding in the time domain, matched filtering of the pulsar signal to extract the signal from the noise and cross-correlation of the pulsar signal with the pulsar “template” (for estimation of TOA) for processing the recordings from two radio telescopes of the studied pulsar signal B0329+54. A rough estimate of TOA measurement accuracy, or pulsar timing accuracy, we have estimated according to the approach proposed by Loriner [1] with the width of the cross-correlation peak at the 0.9 level. Numerical results were obtained in these previous papers on the relationship between the number of accumulated pulses at the folding and the SNR, and the accuracy of the TOA estimation as a function of the SNR, for both types of pulsar signal recordings from a radio telescope with a very large antenna around 6000 sq. m (Westerbork), and a much smaller antenna about 500 sq.m (Dwingeloo). From our results and the results obtained in the project, it followed that the accuracy of the TOA estimation by time / distance is extremely insufficient for real-time navigation of an aircraft. These studies showed that the proposed concept of a pulsar navigation system with a small antenna working in real time for the needs of aircraft (transport) navigation, using a mirror antenna and standard pulsar timing, is not adequate. Because the energy from a pulsar received within a few minutes, using a small antenna, is insufficient to perform a quality pulsar timing, guaranteeing the required accuracy of the TOA measurement.
This challenge led us in the present article to think about what known statistical methods and signal processing approaches can be used, to maximize as much as possible the receiving pulsar energy at the input of the small antenna device, and to speed up and improve the quality of pulsar timing, for a few minutes. Since we work in the field of radar signal processing, which is designed to work in real time, we propose to use its concepts, approaches and algorithms. Most of the known radar signal processing tasks, such as detection, signal parameter estimation, velocity distance and phase estimation, in the statistical theory are solved and developed for different models of signal and interference.
We first checked in the present paper (in chapters 4 and 5) the quality of the signal algorithms proposed and developed by us for assessing the accuracy of TOA estimation presented in previous publications. The verification consisted of evaluating their proximity to the potential CLBR accuracy estimates, at the same SNR ratios as for the recorded pulsar signals. In Chapter 4, a potential TOA estimate was proposed to be calculated as a Crammer-Law Low Bound (CRLB-estimate), the calculation of which also requires an estimate of the SNR level and the parameters of the profile of the pulsar B0329+54, taken from the European pulsar database. From the results obtained in chapter 5, it follows that the rough estimates of TOA obtained by us, at high levels of SNR are close to the potential CRLB- estimates. In order to maximize as much as possible, the received pulsar energy at the input of a small antenna device, we propose in the present paper (Chapter 6) to the used radio astronomy pulsar timing approach, including antenna, receiver gain, dedispersion, and folding, to add all possible radar approaches for receiving signals together with their algorithms, adapted to the specifics of pulsar signals to be used in the processing in the antenna system.
The most used and well-known radar mathematical approach for the description of signal processing in the antenna arrays is the vector representation of e/m field of the pulsar signal for the needs of the spatial processing in the linear antenna arrays. This presentation of e/m pulsar single signal field describes it with a very large number of points, in contrast to the single point description of the pulsar signal field used in mirror antennas. It allows more effective optimal spatial (with antenna arrays) processing of the pulsar signals, and increases the SNR of the received spatial energy. Following this approach, we propose in this paper the simultaneous use of several antenna arrays, matched filtering with the shape of the pulsar signal in each channel of the linear antenna array and multi-frequency simultaneous reception of the pulsar signals in these antenna arrays from the same pulsar. All these signal processing jointly increase the maximum possible SNR ratio at the output of the antenna system. In order to satisfy the requirements for real-time operation in the estimation of the parameters of pulsar signals (TOA and Doppler shift), we propose to use a multi-channel (parallel) approach. It allows at the same time to perform the estimation of the Doppler velocity of the vehicle in different frequency channels of the measurement frequency interval. In each of these frequency channels, TOA estimation is performed simultaneously in time channels of the measured time interval.
The proposed ground navigation algorithm (including antenna and pulsar timing) is developed under the following constraints. Appropriate measures have been taken for stabilization of the received phase pulsar pulses in the navigation receiver, as is done in radars (systems of phase autoadjustment), to measure the speed of a moving vehicle. The algorithm for dedispersion of pulsar signals is assumed to be available in the spatial-time processing channels and is not considered separately in the proposed article.
Question
Use the professional astronomical language for description of pulsar properties.
Answer:
From your point of view, you are correct.
We consider the processing of pulsar signals and their characteristics, in antenna and pulsar timing, and because of this we use the language of statistical signal processing theory.
Please see the attachment.

Round 2
Reviewer 1 Report
The paper has been improved. However, the final conclusion section about the significance of the results should described. On the references, NASA has published some related paper on the pulsar navigation, so the authors might find them to take as references. Moreover, fig.3 should be not an original plot, thus its original reference should be added, or a new fig.3 is plotted; and, on the other figures, authors should double check their references if cited from the journals or books.
The english expression on the math expression part is not so clear, so authors might improve it.
Author Response
COVER LETTER
with authors’ replies for Reviewer 1.(Round 2)
Questions:
The paper has been improved.
- However the final conclusion section about the significance of the results should described.
- On the references, NASA has published some related paper on the pulsar navigation, so the authors might find them to take as references.
Question 2: On the references, NASA has published some related paper on the pulsar navigation, so the authors might find them to take as references.
Answer: Adding this text in lines 268 to 275
NICER and SEXTANT demonstrate XNAV pulsar navigation system that may be used on Artemis [38]. According to the author “NASA's Neutron star Interior Composition Explorer (NICER) instrument has been operating on the International Space Station (ISS) for the past several years. A technology demonstration known as SEXTANT is using these observations as a testbed for future pulsar-based navigation. SEXTANT turns the detector's pulsar timings—especially timing measurements of rapidly spinning millisecond pulsars—into a next-generation autonomous navigation system for precise positioning throughout the solar system”. The U.S. Navy and Department of Defense take cyber threats to technology infrastructure seriously [39]. “Commercial GPS jammers are now readily available on the Internet. A chain reaction of space debris, known as an ablation cascade, can interfere with GPS capability or a strong Earth-centered solar storm. As alert levels rise and military leaders try to assess the situation, ships at sea must somehow get an accurate fix on their position without the use of GPS, and has just resumed training officers in the lost art of celestial navigation.”
Question 3.: Moreover, fig.3 should be not an original plot, thus its original reference should be added, or a new fig.3 is plotted; and, on the other figures, authors should double check their references if cited from the journals or books.
Answer: This fig.3 is taken from [37]: Testing Out Pulsar Navigation by Paul Gilster | Jun 12, 2013 | Communications and Navigation | https://www.centauri-dreams.org/2013/06/12/testing-out-pulsar-navigation/
Question 1.: However the final conclusion section about the significance of the results should described.
Answer: Dear Reviewer 1: I am sending to you a corrected version of the paper in which all the introduction, the conclusion and the abstract are revised. We formulated and explained clearer the scientific tasks we are solving with the paper and the achieved new results. In order to further improve the paper and including the new formulations of the solved tasks and results, we made some corrections in the text in all of the chapters. You can see all the corrections below.
- Introduction
This text is deleted from lines 52 to 75
the further development of known algorithms for signal processing of pulsar signals for the needs of radio navigation of ground transport is proposed.
The emergence of such technology for space passive navigation in the future would increase the quality and reliability of services in the field of modern transport logistics.
This technology will be used for navigation of piloted and unmanned transport vehicles - trucks, cars, drones, in the conditions of joint use of modern space satellite radio navigation systems GNSS, such as GPS, GLONASS, and terrestrial radio networks of GSM from 3,4, and 5-th generation [13].
Traditionally, radio telescopes perform maximum accumulation (folding) of the energy of signals from observed pulsars over the course of hours, which contradicts the requirement for real-time operation. For this reason, it is necessary to look for other solutions for receiving and processing pulsar signals in real time. One of them is using the methods of receiving and processing radio signals in radars that work in real time.
In the proposed article, the theory and principles of pulsar space navigation proposed in works [1,2] are used. We take into account that the navigation pulsar system offered by us will not work in the cosmic space, but on the earth's surface or in the air. That made us search for similarities with the principles and theory of GNSS radio navigation that is so widely used at present.
According to the concept of the structure of a pulsar navigation system proposed in works [1,2], it consists of 4 parts: instrumental stage: antennas; timing estimation stage: estimation of pulsar timing parameters; position estimation stage: determination of position; navigation stage: higher levels of navigational information.
Our scientific task in this article is in the field of receiving and signal processing of pulsar signals for the needs of transport navigation, (antennas; timing estimation stage,).
In its place this text is added from lines 52 to 82
The article is devoted to the theoretical aspects regarding the development of an autonomous radio navigation system with a small receiving antenna, using radio signals from pulsars, similar to navigation systems for space navigation. The transport navigation concept requires significantly reducing the observation time (within a few minutes), providing the transport positioning accuracy that is close to potential using B0329+54-type pulsars. The paper shows that in theoretical papers on space and aviation navigation published years ago, systems with small receiving antennas (10 sq.m) using pulsar signals have rather coarse positioning accuracy. Like GNSS systems, they use signals from 4 suitable pulsars (out of 50) to position the object. These pulsars are not uniformly distributed, but are grouped in certain directions (at least 6 clusters can be determined). Such a pulsar is the pulsar B0329+54 researched in the article.
The transport navigation concept requires significantly reducing the observation time (within a few minutes), providing the transport positioning accuracy that is close to potential using B0329+54-type pulsars, with not large antenna. This is the one of the scientific task that is solved in the paper by studying the relationship between the SNR of the receiver output, which depends on the size of the antenna, the type of signal processing, and the magnitude of the TOA accuracy estimate. The second scientific task that is solved in the paper, is the adaptation of all the possible approaches and algorithms suggested in the statistical theory of radars in the suggested signal algorithm for antenna processing and to evaluate the parameters of the TOA and DS pulsar signals, in order to increase the SNR ratio at the receiver output, while preserving the dimensions of the antenna.
In Chapter 3, theoretical estimates of positioning accuracy of space objects previously obtained by Sala's team are as follows: 0.05∙106 m - for 10 minutes of observation; 0.3∙105 m. (50 – 30 km) - for 100 minutes of observation; 01∙105 m – for 400 minutes of observation. Under all other ideal operating conditions of the space navigation system, these values of positioning accuracy are grossly inappropriate for real-time operation. The article proposes real (experimental) rough position estimates (in time only) of the TOA obtained using signal records from pulsar B0329+54. They are in the range from 24 km to 1 km with an observation time of 2-3 minutes
From lines 82 to 86 the text is moved to lines 62 - 67
In our research, we use the well-known rougher temporal (not phased) approach to transport navigation that was successfully used in early versions of GNSS, because unlike artificial signals for positioning in GNSS, pulsar signals are not modulated and it is impossible to identify the number of phase cycles (pulsar periods) between two signal positions.
The text between the lines 87 to 99 is removed
Our previous research results are also commented on in the present article in chapter 3. They have been connected with the development of signal algorithms including: folding in the time domain, matched filtering of the pulsar signal to extract the signal from the noise and cross-correlation of the pulsar signal with the pulsar “template” (for estimation of Time-Of-Arrival, TOA) for processing the recordings from two radio telescopes of the studied pulsar signal B0329+54. A rough estimate of TOA measurement accuracy, or pulsar timing accuracy, we have estimated according to the approach proposed by Loriner [1] with the width of the cross-correlation peak at the 0.9 level.
Numerical results were obtained in these previous papers on the relationship between the number of accumulated pulses at the folding and the SNR (Signal-to-Noise Ratio), and the accuracy of the TOA estimation as a function of the SNR, for both types of pulsar signal recordings from a radio telescope with a very large antenna around 7000 sq. m (Westerbork), and a much smaller antenna about 500 sq.m (Dwingeloo).
The text between the lines 95 to 100 is moved to lines 78 -93
This challenge led us in the present article to think about what known statistical methods and signal processing approaches can be used, to maximize as much as possible the receiving pulsar energy at the input of the small antenna device, and to speed up and improve the quality of pulsar timing, for a few minutes. Since we work in the field of radar signal processing, which is designed to work in real time, we propose to use its concepts, approaches and algorithms.
In Chapter 4, a potential TOA estimate was proposed to be calculated as a CRLB-estimate, the calculation of which also requires an estimate of the SNR level and the parameters of the profile of the pulsar B0329+54, taken from the European pulsar database.
The text between the lines 92 to 102 is removed
From our results and the results obtained in the project, it followed that the accuracy of the TOA estimation by time / distance is extremely insufficient for real-time navigation of an aircraft. These studies showed that the proposed concept of a pulsar navigation system with a small antenna (about tens to a several hundred square meter) working in real time for the needs of aircraft (transport) navigation, using a mirror antenna and standard pulsar timing, is not adequate. Because the energy from a pulsar received within a few minutes, using a small antenna, is insufficient to perform a quality pulsar timing, guaranteeing the required accuracy of the TOA measurement.
Most of the known radar signal processing tasks, such as detection, signal parameter estimation, velocity distance and phase estimation, in the statistical theory are solved and developed for different models of signal and interference.
The text between the lines 103 to 115 is removed
In order to maximize as much as possible, the received pulsar energy at the input of a small antenna device, we propose in the present paper (Chapter 6) to the used radio astronomy pulsar timing approach, including antenna, receiver gain, dedispersion, and folding, to add all possible radar approaches for receiving signals together with their algorithms, adapted to the specifics of pulsar signals to be used in the processing in the antenna system.
The most used and well-known radar mathematical approach for the description of signal processing in the antenna arrays is the vector representation of e/m field of the pulsar signal for the needs of the spatial processing in the linear antenna arrays. This presentation of e/m pulsar single signal field describes it with a very large number of points, in contrast to the single point description of the pulsar signal field used in mirror antennas. It allows more effective optimal spatial (with antenna arrays) processing of the pulsar signals, and increases the SNR of the received spatial energy.
In its place this text is added between the lines 107 to 127
From the results obtained in chapter 5, it follows that the rough estimates of TOA obtained by us, at high levels of SNR are close to the potential CRLB- estimates. They approach the CRLB estimates, which for the pulsar B0329+54 are from 1 to 10 nm, with large SNR (20-30dB). The article states that it is necessary to look for opportunities to significantly reduce the size of the antenna system compared to those used in the experiment (500m2 and 7000m2.) while ensuring a high level of SNR.
In Chapter 6, the paper suggests all the different approaches proposed in statistical radar theory to increase the receiver output SNR ratio while saving antenna dimensions. These approaches include space-time processing of signals in an antenna array, multi-beam reception of the various broadband pulsar signals for navigation (not less than 4), simultaneous reception of signals from one pulsar at different observation frequencies.
Following this approach, we propose in this paper the simultaneous use of several antenna arrays, matched filtering with the shape of the pulsar signal in each channel of the linear antenna array and multi-frequency simultaneous reception of the pulsar signals in these antenna arrays from the same pulsar. All these signal processing jointly increase the maximum possible SNR ratio at the output of the antenna system.
In order to satisfy the requirements for real-time operation in the estimation of the parameters of pulsar signals (TOA and Doppler shift), we propose to use a multi-channel (parallel) approach. It allows at the same time to perform the estimation of the Doppler velocity of the vehicle in different frequency channels of the measurement frequency interval. In each of these frequency channels, TOA estimation is performed simultaneously in time channels of the measured time interval.
The proposed estimates of positioning accuracy (TOA only, no phase) in an autonomous pulsar vehicle navigation system would only be suitable for navigation of large vehicles (sea, air or land) that do not require accurate navigation at sea, air, desert. Large-sized antennas with an area of tens of square meters to hundreds of square meters can be installed in such vehicles.
This text is moved to lines 629 – 634
The proposed ground navigation algorithm (including antenna and pulsar timing) is developed under the following constraints. Appropriate measures have been taken for stabilization of the received phase pulsar pulses in the navigation receiver, as is done in radars (systems of phase auto adjustment), to measure the speed of a moving vehicle. The algorithm for dedispersion of pulsar signals is assumed to be available in the spatial-time processing channels and is not considered separately in the proposed article.
- Conclusion
Instead of this text from lines 832 to 888
The article analyzes the potential possibility of implementing a transport (ground, air, sea) navigation system using pulsar signals. Methods are proposed for calculating theoretical and experimental estimates (rough estimate and CRLB-estimate) of the accuracy of TOA of pulsar signals. The experimental estimates of the TOA accuracy, both coarse and CRLB, were obtained by computer processing of signal records from the pulsar B0329+54 obtained using the Westerbork and Dwingeloo radio telescopes (Netherlands).
It is shown that the use of the traditional concept of the operation and construction of the single-channel radio receiver, a small mirror antenna (about 10 – 100 sq .m) a highly sensitive receiver, and signal processing with signal accumulation does not allow to achieve the necessary distance measurement accuracy (from TOA) in a transport navigation system operating in real-time mode.
The most difficult problem to solve for transport navigation with pulsars is the search for new ways to quickly accumulate the necessary SNR of the received pulsar signals at the input of the pulsar timing TOA estimation, when using small-sized antennas about (100 m2 or several 100 m2 ).
Real estimates, or experimental rough estimates of the TOA accuracy (according to the results presented in Tables 1 and 2), were obtained for the observed pulsar B0329+54, from the radio equipment of the radio telescopes at the Westerbork and Dwingeloo observatories in the Netherlands.
They are within the limits ~24 km to ~1 km (2.3805 104 - 0.9405 103m.), and were obtained during an observation time of about 2-3 minutes, to accumulate sufficient energy to obtain SNR ratios of 20-30 dB for TOA estimation. Тhe observation time according to Tables 1 and 2 is determined by the number of foldings of the pulsar signals (125-150), as the repetition period of the pulsar B0329+54 signal is T=0.71452 s.
Anyway, if we want to use on the ground the transport radio pulsar navigation system operating in real-time mode and providing rough accuracy of several nautical miles (1 to 10) for positioning and navigation, realized through the evaluation of TOA, it is necessary to look for possibilities of significantly reducing the size of the antenna system compared to the existing ones (500m and 7000m.).
A generalized functional structure of signal processing in a transport navigation system is proposed. This proposed algorithm contains joint space-time processing in the antenna array of vector signals emitted by each observed pulsar and temporal processing of a quasi-deterministic pulsar signal accumulated in the antenna array for multi-channel estimation of its parameters, TOA and Doppler frequency.
In this chapter, we tried to apply a mathematical apparatus for vector description of signals to pulsar signals received at the input of a linear antenna array. The vector description of the received signal allowed us to describe spatio-temporal processing of the pulsar signal in the linear antenna array.
First, in each element of the antenna array, the accumulation (folding) of all pulsar pulses received at the input of the antenna array is performed. The next stage of processing is summation of the accumulated signal on all elements of the linear antenna array. Furthermore, the SNR can also be increased by receiving broadband pulsar signals at several different observation frequencies, since pulsars are known to emit signals over a very wide radio frequency range. Therefore, in the same beam of the antenna array of the receiver, a sufficient number of signals from the same pulsar at different observation frequencies can be simultaneously received.
All these stages of processing allow to significantly increase the SNR of the accumulated pulse at the input of the pulsar timing TOA estimation, to the needed SNR ratios of 20-30 dB for TOA estimation [17,20,21], depending on the antenna size.
The proposed signal processing algorithm for TOA and Doppler frequency shift estimation is suitable for real-time operation because it can be easily implemented in parallel structures (for example VLSI). The proposed functional structure of signal processing and its algorithm can be successfully used for any type of ground, sea and slow air transport (flying cars, airships, balloons).
The mathematical apparatus suggested below could be used for the theoretical evaluation of the SNR ratio for TOA estimation, depending on the antenna size, the pulsar signal parameters, and the specificity of all stages of signal processing.
This text is added from lines 832 to 893
The article is devoted to the theoretical aspects regarding the development of an autonomous radio navigation system with a small receiving antenna, using radio signals from pulsars, similar to navigation systems for space navigation. The paper shows that in theoretical papers on space and aviation navigation published years ago, systems with small receiving antennas (10 sq.m) using pulsar signals have rather coarse positioning accuracy. Like GNSS systems, they use signals from 4 suitable pulsars (out of 50) to position the object. These pulsars are not uniformly distributed, but are grouped in certain directions (at least 6 clusters can be determined). The best pulsars for solving the problem of positioning ( the ambiguity of resolution) are pulsars with large repetition periods of the emitted pulses. It turns out that they are also the most powerful pulsars in terms of the SNR of the emitted pulses. Such a pulsar is the pulsar B0329+54 researched in the article. Theoretical estimates of positioning accuracy of space objects previously obtained by Sala's team are as follows: 0.05∙106 m - for 10 minutes of observation; 0.3∙105 m. (50 – 30 km) - for 100 minutes of observation; 01∙105 m – for 400 minutes of observation. Under all other ideal operating conditions of the space navigation system, these values of positioning accuracy are grossly inappropriate for real-time operation.
When using small antennas (with an area of up to tens of square meters) for pulsar navigation, the energy of the pulsar signals received within a few minutes is extremely insufficient to obtain the required level of SNR at the output of the receiver to form TOA estimation, ensuring positioning accuracy up to tens of kilometers. It follows that according to the theory of space navigation by pulsars, no better positioning accuracy can be expected in pulsar navigation of ground transport when using very small antennas with an area of 10 sq.m. If the size of the receiving antenna is increased to practically acceptable sizes (from tens of square meters to hundreds of square meters), then using only pulse folding, we can increase the SNR by several decibels (5-10 dB), which will reduce the observation time to several tens of minutes, but will not improve the positioning accuracy, which will be tens of kilometers.
The transport navigation concept requires significantly reducing the observation time (within a few minutes), providing the transport positioning accuracy that is close to potential using B0329+54-type pulsars.
This is the main scientific task that is solved in the paper by studying the relationship between the SNR of the receiver output, which depends on the size of the antenna, the type of signal processing, and the magnitude of the TOA accuracy estimate.
The article proposes real (experimental) rough position estimates (in time only) of the TOA obtained using signal records from pulsar B0329+54. These recordings were made by the radio telescopes with antenna sizes (500m2 – 7000m2) at the Westerbork and Dwingeloo observatories in the Netherlands. The TOA estimates presented in the paper were obtained at high SNR values (20-30dB). They are in the range from 24 km to 1 km with an observation time of 2-3 minutes. They approach the CRLB estimates, which for the pulsar B0329+54 are from 1 to 10 nm. The article states that it is necessary to look for opportunities to significantly reduce the size of the antenna system compared to those used in the experiment (500m2 and 7000m2.) while ensuring a high level of SNR.
The paper also suggests all the different approaches proposed in statistical radar theory to increase the receiver output SNR ratio while saving antenna dimensions. These approaches include space-time processing of signals in an antenna array, multi-beam reception of the various broadband pulsar signals for navigation (not less than 4), simultaneous reception of signals from one pulsar at different observation frequencies. The space-time processing of the pulse signal for each received beam performs the accumulation of all pulses in each antenna array element and the summation of the accumulated signal over all array elements, followed as well as with coherent filtering of the pulse signal in multi-channel processing for estimation of the TOA and Doppler frequency shift.
In our opinion, all these processing steps can significantly increase the SNR of the accumulated pulse to the required levels of 20-30 dB for rough TOA estimation [17,20,21], depending on the chosen antenna size. The proposed algorithms for space-time processing of the signal in the antenna array and for multi-channel estimation of the TOA and Doppler frequency shift are suitable for real-time operation, as they can be implemented in the form of parallel computing structures (for example, VLSI).
The proposed estimates of positioning accuracy (TOA only, no phase) in an autonomous pulsar vehicle navigation system would only be suitable for navigation of large vehicles (sea, air or land) that do not require accurate navigation at sea, air, desert. Large-sized antennas with an area of tens of square meters to hundreds of square meters can be installed in such vehicles.
- Navigation with Pulsar Radio Signals
This text is removed from lines 155-161
The pulsar has a colossal mass and high temperature on its surface, and the rotating magnetic field creates an electric field of enormous intensity capable of accelerating protons and electrons almost to the speed of light. All these charged particles moving around the pulsar are trapped by its colossal magnetic field. And only within a small solid angle near the magnetic axis, they can be released (neutron stars have the strongest magnetic fields in the universe, reaching 1010-1014 gauss) [1]. It is these streams of charged particles that are the source of radio emission.
This text is added from lines 162 to 167
These pulsars are not uniformly distributed, but are grouped in certain directions (at least 6 clusters can be determined). The best pulsars for solving the problem of positioning (the ambiguity of resolution) are pulsars with large repetition periods of the emitted pulses. It turns out that they are also the most powerful pulsars in terms of the SNR of the emitted pulses. Such a pulsar is the pulsar B0329+54 researched in the article.
- 3. Overview of Potential Navigational Accuracy by using Pulsar Radio Signals
This text is removed from lines 314 - 322
It is known that the accuracy of estimating the coordinates of space objects depends to the greatest extent on the accuracy of determining the delay time of the signal from the three or more pulsars relative to the synchronizing signal of the navigation system. The delay time of the pulsar signal (TOA) is necessary to determine the distance to the pulsars and solve the navigation task.
Similar studies on the navigational accuracy of cosmic objects with small antennas (up to 10 sq. m) using radio pulsars were conducted in 2000 [2]. In [2], one possible algorithm for navigating several pulsar signals already extracted from receiver noise is considered in using small antennas up to 10 m2.
It is known that the accuracy of estimating the coordinates of space objects depends to the greatest extent on the accuracy of determining the delay time of the signal from the three or more pulsars relative to the synchronizing signal of the navigation system. The delay time of the pulsar signal (TOA) is necessary to determine the distance to the pulsars and solve the navigation task.
This text is added in its place from lines 314 to 328
The paper shows that in theoretical papers on space and aviation navigation published years ago, systems with small receiving antennas (10 sq.m) using pulsar signals have rather coarse positioning accuracy.
Like GNSS systems, they use signals from 4 suitable pulsars (out of 50) to position the object. These pulsars are not uniformly distributed, but are grouped in certain directions (at least 6 clusters can be determined). The best pulsars for solving the problem of positioning ( the ambiguity of resolution) are pulsars with large repetition periods of the emitted pulses. It turns out that they are also the most powerful pulsars in terms of the SNR of the emitted pulses. Such a pulsar is the pulsar B0329+54 researched in the article.
Theoretical estimates of positioning accuracy of space objects previously obtained by Sala's team are as follows: 0.05∙106 m - for 10 minutes of observation; 0.3∙105 m. (50 – 30 km) - for 100 minutes of observation; 01∙105m – for 400 minutes of observation. Under all other ideal operating conditions of the space navigation system, these values of positioning accuracy are grossly inappropriate for real-time operation.
This text is added from lines 333 to 342
The articles [15, 17] propose real (experimental) rough position estimates (in time only) of the TOA obtained using signal records from pulsar B0329+54. These recordings were made by the radio telescopes with antenna sizes (500m2 – 7000m2) at the Westerbork and Dwingeloo observatories in the Netherlands, (Figure 9 and Figure10). The TOA estimates presented in the paper were obtained at high SNR values (20-30dB). They are in the range from 24 km to 1 km with an observation time of 2-3 minutes. They approach the CRLB estimates, which for the pulsar B0329+54 are from 1 to 10 nm. These estimates were calculated after signal and statistical processing in the MATLAB environment. This is grossly inadequate for aircraft navigation, and is perhaps more suitable for marine navigation.
This text is added from lines 361 to 363
For system temperature T= 15-150 K accuracy 0,2∙105 m - for 35 minutes of observation, and 0,25 105 m - for 865 minutes of observation
This text is removed from lines 365-370
The recent emergence of large transport drones carrying cargo hundreds of kilometers outside urban and sea environments (beyond 5G) and also the emergence of many new receivers for 5G needs (e.g. Starlink receivers) motivates us to revisit this topic. The technological solution to the task of transport navigation can also be found using the theory and practice of radio photonics, radio-optical antenna arrays and digital coherent-optical processors.
- Numerical Results for TOA Accuracy Estimates After Processing Pulsar Signals from the Pulsar B0329+54
Instead of this text from lines 566 -576
Anyway, if we want to use on the ground the transport radio pulsar navigation system operating in real-time mode and providing rough accuracy of several nautical miles (1 to 10) for positioning and navigation, realized through the evaluation of TOA, it is necessary to look for possibilities of significantly reducing the size of the antenna system compared to the existing ones (500m and 7000m.)
According to [32, 28], the statistical theory of radar offers many other approaches, methods and optimal algorithms, allowing to significantly increase the SNR of the input radio signal against the background of white Gaussian noise, at the input of the pulsar timing, when using small-sized antennas. These are matched coherent accumulation of radio signal packets, joint spatial and temporal processing of radio signals (STAP), aperture synthesis of a moving transport (ground, air, sea) navigation system..
This text is added in its place in lines 565 to 569
The article states that it is necessary to look for opportunities to significantly reduce the size of the antenna system compared to those used in the experiment (500m2 and 7000m2.) while ensuring a high level of SNR. The paper also suggests all the different approaches proposed in statistical radar theory to increase the receiver output SNR ratio while saving antenna dimensions.
- Optimal multichannel space-time algorithm for estimation of TOA and Doppler frequency of vector pulsar signals
This text is removed from lines 584 to 586
In this chapter, a radar approach is proposed for the synthesis of optimal processing in antenna arrays of a series of vector signals from pulsars for the needs of transport navigation in the real-time mode. Using the approach of vector time functions in the synthesis of optimal signal processing systems allows to simplify the complex theory of optimal processing of random electromagnetic fields. This field-to-vector transformation agrees well with the real electromagnetic field discretization performed in the multi-element antennas (antenna arrays).
As already mentioned, the solving the most difficult problem for transport navigation with pulsars is in the search for new ways to quickly accumulate the necessary SNR of the received pulsar signals at the input of the pulsar timing TOA estimation, when using small-sized antennas about (100 m2 or several 100 m2 ). The antenna size of the navigation device depends on the accuracy required for navigation, to accumulate sufficient energy to obtain SNR ratios of 20-30 dB for TOA estimation [17,20,21].
In its place this text is added in lines 589-593
That is why the main scientific task that is solved in the paper, is the adaptation of all the possible approaches and algorithms suggested in the statistical theory of radars in the suggested signal algorithm for antenna processing and to evaluate the parameters of the TOA and DS pulsar signals, in order to increase the SNR ratio at the receiver output, while preserving the dimensions of the antenna.

Reviewer 4 Report
I did't see the significant changes in text in order to accept it for publication.
It is not important.
Author Response
COVER LETTER
with authors’ replies for Reviewer 4.(Round 2)
Questions:
Question 1.: The introduction and conclusion must be improved.
Answer: Dear Reviewer 4: I am sending to you a corrected version of the paper in which all the introduction, the conclusion and the abstract are revised. We formulated and explained clearer the scientific tasks we are solving with the paper and the achieved new results. In order to further improve the paper and including the new formulations of the solved tasks and results, we made some corrections in the text in all of the chapters. You can see all the corrections below.
- Introduction
This text is deleted from lines 52 to 75
the further development of known algorithms for signal processing of pulsar signals for the needs of radio navigation of ground transport is proposed.
The emergence of such technology for space passive navigation in the future would increase the quality and reliability of services in the field of modern transport logistics.
This technology will be used for navigation of piloted and unmanned transport vehicles - trucks, cars, drones, in the conditions of joint use of modern space satellite radio navigation systems GNSS, such as GPS, GLONASS, and terrestrial radio networks of GSM from 3,4, and 5-th generation [13].
Traditionally, radio telescopes perform maximum accumulation (folding) of the energy of signals from observed pulsars over the course of hours, which contradicts the requirement for real-time operation. For this reason, it is necessary to look for other solutions for receiving and processing pulsar signals in real time. One of them is using the methods of receiving and processing radio signals in radars that work in real time.
In the proposed article, the theory and principles of pulsar space navigation proposed in works [1,2] are used. We take into account that the navigation pulsar system offered by us will not work in the cosmic space, but on the earth's surface or in the air. That made us search for similarities with the principles and theory of GNSS radio navigation that is so widely used at present.
According to the concept of the structure of a pulsar navigation system proposed in works [1,2], it consists of 4 parts: instrumental stage: antennas; timing estimation stage: estimation of pulsar timing parameters; position estimation stage: determination of position; navigation stage: higher levels of navigational information.
Our scientific task in this article is in the field of receiving and signal processing of pulsar signals for the needs of transport navigation, (antennas; timing estimation stage,).
In its place this text is added from lines 52 to 82
The article is devoted to the theoretical aspects regarding the development of an autonomous radio navigation system with a small receiving antenna, using radio signals from pulsars, similar to navigation systems for space navigation. The transport navigation concept requires significantly reducing the observation time (within a few minutes), providing the transport positioning accuracy that is close to potential using B0329+54-type pulsars. The paper shows that in theoretical papers on space and aviation navigation published years ago, systems with small receiving antennas (10 sq.m) using pulsar signals have rather coarse positioning accuracy. Like GNSS systems, they use signals from 4 suitable pulsars (out of 50) to position the object. These pulsars are not uniformly distributed, but are grouped in certain directions (at least 6 clusters can be determined). Such a pulsar is the pulsar B0329+54 researched in the article.
The transport navigation concept requires significantly reducing the observation time (within a few minutes), providing the transport positioning accuracy that is close to potential using B0329+54-type pulsars, with not large antenna. This is the one of the scientific task that is solved in the paper by studying the relationship between the SNR of the receiver output, which depends on the size of the antenna, the type of signal processing, and the magnitude of the TOA accuracy estimate. The second scientific task that is solved in the paper, is the adaptation of all the possible approaches and algorithms suggested in the statistical theory of radars in the suggested signal algorithm for antenna processing and to evaluate the parameters of the TOA and DS pulsar signals, in order to increase the SNR ratio at the receiver output, while preserving the dimensions of the antenna.
In Chapter 3, theoretical estimates of positioning accuracy of space objects previously obtained by Sala's team are as follows: 0.05∙106 m - for 10 minutes of observation; 0.3∙105 m. (50 – 30 km) - for 100 minutes of observation; 01∙105 m – for 400 minutes of observation. Under all other ideal operating conditions of the space navigation system, these values of positioning accuracy are grossly inappropriate for real-time operation. The article proposes real (experimental) rough position estimates (in time only) of the TOA obtained using signal records from pulsar B0329+54. They are in the range from 24 km to 1 km with an observation time of 2-3 minutes
From lines 82 to 86 the text is moved to lines 62 - 67
In our research, we use the well-known rougher temporal (not phased) approach to transport navigation that was successfully used in early versions of GNSS, because unlike artificial signals for positioning in GNSS, pulsar signals are not modulated and it is impossible to identify the number of phase cycles (pulsar periods) between two signal positions.
The text between the lines 87 to 99 is removed
Our previous research results are also commented on in the present article in chapter 3. They have been connected with the development of signal algorithms including: folding in the time domain, matched filtering of the pulsar signal to extract the signal from the noise and cross-correlation of the pulsar signal with the pulsar “template” (for estimation of Time-Of-Arrival, TOA) for processing the recordings from two radio telescopes of the studied pulsar signal B0329+54. A rough estimate of TOA measurement accuracy, or pulsar timing accuracy, we have estimated according to the approach proposed by Loriner [1] with the width of the cross-correlation peak at the 0.9 level.
Numerical results were obtained in these previous papers on the relationship between the number of accumulated pulses at the folding and the SNR (Signal-to-Noise Ratio), and the accuracy of the TOA estimation as a function of the SNR, for both types of pulsar signal recordings from a radio telescope with a very large antenna around 7000 sq. m (Westerbork), and a much smaller antenna about 500 sq.m (Dwingeloo).
The text between the lines 95 to 100 is moved to lines 78 -93
This challenge led us in the present article to think about what known statistical methods and signal processing approaches can be used, to maximize as much as possible the receiving pulsar energy at the input of the small antenna device, and to speed up and improve the quality of pulsar timing, for a few minutes. Since we work in the field of radar signal processing, which is designed to work in real time, we propose to use its concepts, approaches and algorithms.
In Chapter 4, a potential TOA estimate was proposed to be calculated as a CRLB-estimate, the calculation of which also requires an estimate of the SNR level and the parameters of the profile of the pulsar B0329+54, taken from the European pulsar database.
The text between the lines 92 to 102 is removed
From our results and the results obtained in the project, it followed that the accuracy of the TOA estimation by time / distance is extremely insufficient for real-time navigation of an aircraft. These studies showed that the proposed concept of a pulsar navigation system with a small antenna (about tens to a several hundred square meter) working in real time for the needs of aircraft (transport) navigation, using a mirror antenna and standard pulsar timing, is not adequate. Because the energy from a pulsar received within a few minutes, using a small antenna, is insufficient to perform a quality pulsar timing, guaranteeing the required accuracy of the TOA measurement.
Most of the known radar signal processing tasks, such as detection, signal parameter estimation, velocity distance and phase estimation, in the statistical theory are solved and developed for different models of signal and interference.
The text between the lines 103 to 115 is removed
In order to maximize as much as possible, the received pulsar energy at the input of a small antenna device, we propose in the present paper (Chapter 6) to the used radio astronomy pulsar timing approach, including antenna, receiver gain, dedispersion, and folding, to add all possible radar approaches for receiving signals together with their algorithms, adapted to the specifics of pulsar signals to be used in the processing in the antenna system.
The most used and well-known radar mathematical approach for the description of signal processing in the antenna arrays is the vector representation of e/m field of the pulsar signal for the needs of the spatial processing in the linear antenna arrays. This presentation of e/m pulsar single signal field describes it with a very large number of points, in contrast to the single point description of the pulsar signal field used in mirror antennas. It allows more effective optimal spatial (with antenna arrays) processing of the pulsar signals, and increases the SNR of the received spatial energy.
In its place this text is added between the lines 107 to 127
From the results obtained in chapter 5, it follows that the rough estimates of TOA obtained by us, at high levels of SNR are close to the potential CRLB- estimates. They approach the CRLB estimates, which for the pulsar B0329+54 are from 1 to 10 nm, with large SNR (20-30dB). The article states that it is necessary to look for opportunities to significantly reduce the size of the antenna system compared to those used in the experiment (500m2 and 7000m2.) while ensuring a high level of SNR.
In Chapter 6, the paper suggests all the different approaches proposed in statistical radar theory to increase the receiver output SNR ratio while saving antenna dimensions. These approaches include space-time processing of signals in an antenna array, multi-beam reception of the various broadband pulsar signals for navigation (not less than 4), simultaneous reception of signals from one pulsar at different observation frequencies.
Following this approach, we propose in this paper the simultaneous use of several antenna arrays, matched filtering with the shape of the pulsar signal in each channel of the linear antenna array and multi-frequency simultaneous reception of the pulsar signals in these antenna arrays from the same pulsar. All these signal processing jointly increase the maximum possible SNR ratio at the output of the antenna system.
In order to satisfy the requirements for real-time operation in the estimation of the parameters of pulsar signals (TOA and Doppler shift), we propose to use a multi-channel (parallel) approach. It allows at the same time to perform the estimation of the Doppler velocity of the vehicle in different frequency channels of the measurement frequency interval. In each of these frequency channels, TOA estimation is performed simultaneously in time channels of the measured time interval.
The proposed estimates of positioning accuracy (TOA only, no phase) in an autonomous pulsar vehicle navigation system would only be suitable for navigation of large vehicles (sea, air or land) that do not require accurate navigation at sea, air, desert. Large-sized antennas with an area of tens of square meters to hundreds of square meters can be installed in such vehicles.
This text is moved to lines 629 – 634
The proposed ground navigation algorithm (including antenna and pulsar timing) is developed under the following constraints. Appropriate measures have been taken for stabilization of the received phase pulsar pulses in the navigation receiver, as is done in radars (systems of phase auto adjustment), to measure the speed of a moving vehicle. The algorithm for dedispersion of pulsar signals is assumed to be available in the spatial-time processing channels and is not considered separately in the proposed article.
- Conclusion
Instead of this text from lines 832 to 888
The article analyzes the potential possibility of implementing a transport (ground, air, sea) navigation system using pulsar signals. Methods are proposed for calculating theoretical and experimental estimates (rough estimate and CRLB-estimate) of the accuracy of TOA of pulsar signals. The experimental estimates of the TOA accuracy, both coarse and CRLB, were obtained by computer processing of signal records from the pulsar B0329+54 obtained using the Westerbork and Dwingeloo radio telescopes (Netherlands).
It is shown that the use of the traditional concept of the operation and construction of the single-channel radio receiver, a small mirror antenna (about 10 – 100 sq .m) a highly sensitive receiver, and signal processing with signal accumulation does not allow to achieve the necessary distance measurement accuracy (from TOA) in a transport navigation system operating in real-time mode.
The most difficult problem to solve for transport navigation with pulsars is the search for new ways to quickly accumulate the necessary SNR of the received pulsar signals at the input of the pulsar timing TOA estimation, when using small-sized antennas about (100 m2 or several 100 m2 ).
Real estimates, or experimental rough estimates of the TOA accuracy (according to the results presented in Tables 1 and 2), were obtained for the observed pulsar B0329+54, from the radio equipment of the radio telescopes at the Westerbork and Dwingeloo observatories in the Netherlands.
They are within the limits ~24 km to ~1 km (2.3805 104 - 0.9405 103m.), and were obtained during an observation time of about 2-3 minutes, to accumulate sufficient energy to obtain SNR ratios of 20-30 dB for TOA estimation. Тhe observation time according to Tables 1 and 2 is determined by the number of foldings of the pulsar signals (125-150), as the repetition period of the pulsar B0329+54 signal is T=0.71452 s.
Anyway, if we want to use on the ground the transport radio pulsar navigation system operating in real-time mode and providing rough accuracy of several nautical miles (1 to 10) for positioning and navigation, realized through the evaluation of TOA, it is necessary to look for possibilities of significantly reducing the size of the antenna system compared to the existing ones (500m and 7000m.).
A generalized functional structure of signal processing in a transport navigation system is proposed. This proposed algorithm contains joint space-time processing in the antenna array of vector signals emitted by each observed pulsar and temporal processing of a quasi-deterministic pulsar signal accumulated in the antenna array for multi-channel estimation of its parameters, TOA and Doppler frequency.
In this chapter, we tried to apply a mathematical apparatus for vector description of signals to pulsar signals received at the input of a linear antenna array. The vector description of the received signal allowed us to describe spatio-temporal processing of the pulsar signal in the linear antenna array.
First, in each element of the antenna array, the accumulation (folding) of all pulsar pulses received at the input of the antenna array is performed. The next stage of processing is summation of the accumulated signal on all elements of the linear antenna array. Furthermore, the SNR can also be increased by receiving broadband pulsar signals at several different observation frequencies, since pulsars are known to emit signals over a very wide radio frequency range. Therefore, in the same beam of the antenna array of the receiver, a sufficient number of signals from the same pulsar at different observation frequencies can be simultaneously received.
All these stages of processing allow to significantly increase the SNR of the accumulated pulse at the input of the pulsar timing TOA estimation, to the needed SNR ratios of 20-30 dB for TOA estimation [17,20,21], depending on the antenna size.
The proposed signal processing algorithm for TOA and Doppler frequency shift estimation is suitable for real-time operation because it can be easily implemented in parallel structures (for example VLSI). The proposed functional structure of signal processing and its algorithm can be successfully used for any type of ground, sea and slow air transport (flying cars, airships, balloons).
The mathematical apparatus suggested below could be used for the theoretical evaluation of the SNR ratio for TOA estimation, depending on the antenna size, the pulsar signal parameters, and the specificity of all stages of signal processing.
This text is added from lines 832 to 893
The article is devoted to the theoretical aspects regarding the development of an autonomous radio navigation system with a small receiving antenna, using radio signals from pulsars, similar to navigation systems for space navigation. The paper shows that in theoretical papers on space and aviation navigation published years ago, systems with small receiving antennas (10 sq.m) using pulsar signals have rather coarse positioning accuracy. Like GNSS systems, they use signals from 4 suitable pulsars (out of 50) to position the object. These pulsars are not uniformly distributed, but are grouped in certain directions (at least 6 clusters can be determined). The best pulsars for solving the problem of positioning ( the ambiguity of resolution) are pulsars with large repetition periods of the emitted pulses. It turns out that they are also the most powerful pulsars in terms of the SNR of the emitted pulses. Such a pulsar is the pulsar B0329+54 researched in the article. Theoretical estimates of positioning accuracy of space objects previously obtained by Sala's team are as follows: 0.05∙106 m - for 10 minutes of observation; 0.3∙105 m. (50 – 30 km) - for 100 minutes of observation; 01∙105 m – for 400 minutes of observation. Under all other ideal operating conditions of the space navigation system, these values of positioning accuracy are grossly inappropriate for real-time operation.
When using small antennas (with an area of up to tens of square meters) for pulsar navigation, the energy of the pulsar signals received within a few minutes is extremely insufficient to obtain the required level of SNR at the output of the receiver to form TOA estimation, ensuring positioning accuracy up to tens of kilometers. It follows that according to the theory of space navigation by pulsars, no better positioning accuracy can be expected in pulsar navigation of ground transport when using very small antennas with an area of 10 sq.m. If the size of the receiving antenna is increased to practically acceptable sizes (from tens of square meters to hundreds of square meters), then using only pulse folding, we can increase the SNR by several decibels (5-10 dB), which will reduce the observation time to several tens of minutes, but will not improve the positioning accuracy, which will be tens of kilometers.
The transport navigation concept requires significantly reducing the observation time (within a few minutes), providing the transport positioning accuracy that is close to potential using B0329+54-type pulsars.
This is the main scientific task that is solved in the paper by studying the relationship between the SNR of the receiver output, which depends on the size of the antenna, the type of signal processing, and the magnitude of the TOA accuracy estimate.
The article proposes real (experimental) rough position estimates (in time only) of the TOA obtained using signal records from pulsar B0329+54. These recordings were made by the radio telescopes with antenna sizes (500m2 – 7000m2) at the Westerbork and Dwingeloo observatories in the Netherlands. The TOA estimates presented in the paper were obtained at high SNR values (20-30dB). They are in the range from 24 km to 1 km with an observation time of 2-3 minutes. They approach the CRLB estimates, which for the pulsar B0329+54 are from 1 to 10 nm. The article states that it is necessary to look for opportunities to significantly reduce the size of the antenna system compared to those used in the experiment (500m2 and 7000m2.) while ensuring a high level of SNR.
The paper also suggests all the different approaches proposed in statistical radar theory to increase the receiver output SNR ratio while saving antenna dimensions. These approaches include space-time processing of signals in an antenna array, multi-beam reception of the various broadband pulsar signals for navigation (not less than 4), simultaneous reception of signals from one pulsar at different observation frequencies. The space-time processing of the pulse signal for each received beam performs the accumulation of all pulses in each antenna array element and the summation of the accumulated signal over all array elements, followed as well as with coherent filtering of the pulse signal in multi-channel processing for estimation of the TOA and Doppler frequency shift.
In our opinion, all these processing steps can significantly increase the SNR of the accumulated pulse to the required levels of 20-30 dB for rough TOA estimation [17,20,21], depending on the chosen antenna size. The proposed algorithms for space-time processing of the signal in the antenna array and for multi-channel estimation of the TOA and Doppler frequency shift are suitable for real-time operation, as they can be implemented in the form of parallel computing structures (for example, VLSI).
The proposed estimates of positioning accuracy (TOA only, no phase) in an autonomous pulsar vehicle navigation system would only be suitable for navigation of large vehicles (sea, air or land) that do not require accurate navigation at sea, air, desert. Large-sized antennas with an area of tens of square meters to hundreds of square meters can be installed in such vehicles.
- Navigation with Pulsar Radio Signals
This text is removed from lines 155-161
The pulsar has a colossal mass and high temperature on its surface, and the rotating magnetic field creates an electric field of enormous intensity capable of accelerating protons and electrons almost to the speed of light. All these charged particles moving around the pulsar are trapped by its colossal magnetic field. And only within a small solid angle near the magnetic axis, they can be released (neutron stars have the strongest magnetic fields in the universe, reaching 1010-1014 gauss) [1]. It is these streams of charged particles that are the source of radio emission.
This text is added from lines 162 to 167
These pulsars are not uniformly distributed, but are grouped in certain directions (at least 6 clusters can be determined). The best pulsars for solving the problem of positioning (the ambiguity of resolution) are pulsars with large repetition periods of the emitted pulses. It turns out that they are also the most powerful pulsars in terms of the SNR of the emitted pulses. Such a pulsar is the pulsar B0329+54 researched in the article.
- 3. Overview of Potential Navigational Accuracy by using Pulsar Radio Signals
This text is removed from lines 314 - 322
It is known that the accuracy of estimating the coordinates of space objects depends to the greatest extent on the accuracy of determining the delay time of the signal from the three or more pulsars relative to the synchronizing signal of the navigation system. The delay time of the pulsar signal (TOA) is necessary to determine the distance to the pulsars and solve the navigation task.
Similar studies on the navigational accuracy of cosmic objects with small antennas (up to 10 sq. m) using radio pulsars were conducted in 2000 [2]. In [2], one possible algorithm for navigating several pulsar signals already extracted from receiver noise is considered in using small antennas up to 10 m2.
It is known that the accuracy of estimating the coordinates of space objects depends to the greatest extent on the accuracy of determining the delay time of the signal from the three or more pulsars relative to the synchronizing signal of the navigation system. The delay time of the pulsar signal (TOA) is necessary to determine the distance to the pulsars and solve the navigation task.
This text is added in its place from lines 314 to 328
The paper shows that in theoretical papers on space and aviation navigation published years ago, systems with small receiving antennas (10 sq.m) using pulsar signals have rather coarse positioning accuracy.
Like GNSS systems, they use signals from 4 suitable pulsars (out of 50) to position the object. These pulsars are not uniformly distributed, but are grouped in certain directions (at least 6 clusters can be determined). The best pulsars for solving the problem of positioning ( the ambiguity of resolution) are pulsars with large repetition periods of the emitted pulses. It turns out that they are also the most powerful pulsars in terms of the SNR of the emitted pulses. Such a pulsar is the pulsar B0329+54 researched in the article.
Theoretical estimates of positioning accuracy of space objects previously obtained by Sala's team are as follows: 0.05∙106 m - for 10 minutes of observation; 0.3∙105 m. (50 – 30 km) - for 100 minutes of observation; 01∙105m – for 400 minutes of observation. Under all other ideal operating conditions of the space navigation system, these values of positioning accuracy are grossly inappropriate for real-time operation.
This text is added from lines 333 to 342
The articles [15, 17] propose real (experimental) rough position estimates (in time only) of the TOA obtained using signal records from pulsar B0329+54. These recordings were made by the radio telescopes with antenna sizes (500m2 – 7000m2) at the Westerbork and Dwingeloo observatories in the Netherlands, (Figure 9 and Figure10). The TOA estimates presented in the paper were obtained at high SNR values (20-30dB). They are in the range from 24 km to 1 km with an observation time of 2-3 minutes. They approach the CRLB estimates, which for the pulsar B0329+54 are from 1 to 10 nm. These estimates were calculated after signal and statistical processing in the MATLAB environment. This is grossly inadequate for aircraft navigation, and is perhaps more suitable for marine navigation.
This text is added from lines 361 to 363
For system temperature T= 15-150 K accuracy 0,2∙105 m - for 35 minutes of observation, and 0,25 105 m - for 865 minutes of observation
This text is removed from lines 365-370
The recent emergence of large transport drones carrying cargo hundreds of kilometers outside urban and sea environments (beyond 5G) and also the emergence of many new receivers for 5G needs (e.g. Starlink receivers) motivates us to revisit this topic. The technological solution to the task of transport navigation can also be found using the theory and practice of radio photonics, radio-optical antenna arrays and digital coherent-optical processors.
- Numerical Results for TOA Accuracy Estimates After Processing Pulsar Signals from the Pulsar B0329+54
Instead of this text from lines 566 -576
Anyway, if we want to use on the ground the transport radio pulsar navigation system operating in real-time mode and providing rough accuracy of several nautical miles (1 to 10) for positioning and navigation, realized through the evaluation of TOA, it is necessary to look for possibilities of significantly reducing the size of the antenna system compared to the existing ones (500m and 7000m.)
According to [32, 28], the statistical theory of radar offers many other approaches, methods and optimal algorithms, allowing to significantly increase the SNR of the input radio signal against the background of white Gaussian noise, at the input of the pulsar timing, when using small-sized antennas. These are matched coherent accumulation of radio signal packets, joint spatial and temporal processing of radio signals (STAP), aperture synthesis of a moving transport (ground, air, sea) navigation system..
This text is added in its place in lines 565 to 569
The article states that it is necessary to look for opportunities to significantly reduce the size of the antenna system compared to those used in the experiment (500m2 and 7000m2.) while ensuring a high level of SNR. The paper also suggests all the different approaches proposed in statistical radar theory to increase the receiver output SNR ratio while saving antenna dimensions.
- Optimal multichannel space-time algorithm for estimation of TOA and Doppler frequency of vector pulsar signals
This text is removed from lines 584 to 586
In this chapter, a radar approach is proposed for the synthesis of optimal processing in antenna arrays of a series of vector signals from pulsars for the needs of transport navigation in the real-time mode. Using the approach of vector time functions in the synthesis of optimal signal processing systems allows to simplify the complex theory of optimal processing of random electromagnetic fields. This field-to-vector transformation agrees well with the real electromagnetic field discretization performed in the multi-element antennas (antenna arrays).
As already mentioned, the solving the most difficult problem for transport navigation with pulsars is in the search for new ways to quickly accumulate the necessary SNR of the received pulsar signals at the input of the pulsar timing TOA estimation, when using small-sized antennas about (100 m2 or several 100 m2 ). The antenna size of the navigation device depends on the accuracy required for navigation, to accumulate sufficient energy to obtain SNR ratios of 20-30 dB for TOA estimation [17,20,21].
In its place this text is added in lines 589-593
That is why the main scientific task that is solved in the paper, is the adaptation of all the possible approaches and algorithms suggested in the statistical theory of radars in the suggested signal algorithm for antenna processing and to evaluate the parameters of the TOA and DS pulsar signals, in order to increase the SNR ratio at the receiver output, while preserving the dimensions of the antenna.

Round 3
Reviewer 1 Report
1) Authors use the pulsar B0329+54 for the research in the article, however, in general,
as known, the Crab pulsar is very well measured by astronomers
with the timing(https://ui.adsabs.harvard.edu/abs/2015MNRAS.446..857L/abstract),
and perfectly described in theoretical model(https://ui.adsabs.harvard.edu/abs/2022Univ....8..628Z/abstract),
so, it may be discussed in the final section, what is the difference if replacing the pulsar B0329+54 by the Crab pulsar,
or the Crab pulsar is also a good candidate for the navigation?
2) The magnetic field of normal pulsar is averagely B-field~=10^12 Gauss (~10^10 to 10^15 Gauss); while for millisecond pulsars their B-field~ = 10^(8-9) Gauass (https://ui.adsabs.harvard.edu/abs/2022PASP..134k4201Z/abstract).
no
Author Response
COVER LETTER
with authors’ replies for Reviewer 1. (Round 3)
Question 1:
Authors use the pulsar B0329+54 for the research in the article, however, in general,
as known, the Crab pulsar is very well measured by astronomers
with the timing (https://ui.adsabs.harvard.edu/abs/2015MNRAS.446..857L/abstract),
and perfectly described in theoretical model (https://ui.adsabs.harvard.edu/abs/2022Univ....8..628Z/abstract),
so, it may be discussed in the final section, what is the difference if replacing the pulsar B0329+54 by the Crab pulsar,
or the Crab pulsar is also a good candidate for the navigation?
Answer:
In Introduction and in Section 3, the texts have been added, which comment that now: US NASA is developing exactly such navigation systems for military purposes, which are located on a spacecraft (the spacecraft Artemis [38]) and use signals from X-Ray pulsars (X-ray pulsar navigation system with Crab pulsar). These navigation systems are intended to replace GPS GNSS in the event of its intentional or unintentional jamming or destruction by military action.
Most X-ray pulsars emit very weak signals. The Crab pulsar (PSR B0531+21) is the most powerful pulsar in the X-ray range with the radiation flux density ~ 9.9 • 10-9 erg cm -2s-1 with photon energy 2-10 keV. Other pulsars in this range are much weaker emitters. It is known that X-ray and gamma-ray pulsars can only be observed by space crafts, because signals from these pulsars are impossible to be detected on the Earth's surface due to their strong absorption by the Earth's atmosphere.
In the event of a magnetic storm, radio jamming or other blocking of GPS GNSS, other GNSS radio navigation systems, as well as blocking of ground navigation systems, the all transport ground and air means of transport (ships, planes) will remain without cosmic navigation and ground support. In these situations, the military XNAV pulsar navigation system will not be available for civilian navigation. But even in the absence of any threats to space and ground navigation and communication systems of the 4th and 5th generation, the presence of a pulsar navigation system will be an additional source of navigational information in certain areas of the earth's surface.
For these two reasons, the development of a terrestrial radio pulsar navigation system for transportation is a very current scientific and applied task that awaits its solution.
„The best pulsars for solving the problem of positioning (the ambiguity of resolution) are pulsars with large repetition periods of the emitted pulses. It turns out that they are also the most powerful pulsars in terms of the SNR of the emitted pulses. Such a pulsar is the pulsar B0329+54 researched in the article“ (see Lines 169-173).
Moreover, in this research we already had real recordings of this pulsar made earlier at the Westbork and Dwingeloo observatories. These real pulsar recordings were used by us earlier when working on the European PulsaPlane project, which investigated the creation of a radio navigation system for aircraft using radio pulsar signals.
As can be seen from the added overview of the articles [40,41], the navigation accuracy (space) depends on both the choice of pulsars and the type of TOA navigation processing. It follows from this overview that there is no particular difference in navigation accuracy depending on the pulsar emission range, radio or X-ray. The high-precision TOA estimation with phase accuracy allows obtaining an accuracy of 3 km up to 15 km or a rough estimate of TOA - of tens and hundreds of kilometers. See Lines from 286 to 289: „It is shown that in the most favorable case using high-precision measurement, the accuracy of measuring the spacecraft coordinates is 3 km up to 15 km. A rough determination of the position of the spacecraft in space is performed with an accuracy of tens and hundreds of kilometers [40].
To obtain the above navigation accuracies, sensor detectors with an area of 1000 sq.cm are used. Observation times in these space X-ray navigation systems are on the order of 1000 to 10000 sec. In contrast, radio pulsars are unsuitable for space navigation, because the signals from them are weak and for this reason they require for their observation antennas with a diameter in the range of 25-100 m. Obtaining a highly accurate TOA estimate requires quite a large number of iterations (observation time) necessary to overcome the uncertainty of the pulsar pulse phase.
In order to reduce the observation time, the article proposes to use a rough estimate of TOA, because it is best suited for real-time operation of a navigation system and for this reason is used exclusively in radars. In this case, solving the phase uncertainty problem is avoided, thus saving computational resources.
In Section 6, an algorithm is proposed to substantially increase the SNR at the TOA meter input, in order to reduce the antenna size and the processing time in real-time operation.
Question 2:
The magnetic field of normal pulsar is averagely B-field~=10^12 Gauss (~10^10 to 10^15 Gauss); while for millisecond pulsars their B-field~ = 10^(8-9) Gauass (https://ui.adsabs.harvard.edu/abs/2022PASP..134k4201Z/abstract).
Answer:
Most X-ray pulsars emit very weak signals. The Crab pulsar (PSR B0531+21) is the most powerful pulsar in the X-ray range with the radiation flux density ~ 9.9 • 10-9 erg cm -2s-1 with photon energy 2-10 keV. Other pulsars in this range are much weaker emitters.
It is known that X-ray and gamma-ray pulsars can only be observed by spacecraft, because signals from these pulsars are impossible to be detected on the Earth's surface due to their strong absorption by the Earth's atmosphere.
On Earth's surface, only radio pulsars can be used for navigation. In this case, more powerful seconds pulsars can be used, which can be observed with smaller antennas. However, these pulsars have a pulse emission period of the order of 1 sec (seconds pulsars). The TOA measurement error for these pulsars is of the order of tens-hundreds of microseconds.
The other group of pulsars are so-called milliseconds pulsars, the pulse repetition period of which is on the order of several milliseconds. However, these pulsars have less radiation power and therefore require antennas with a larger area for observation. However, the measurement error of TOA is less than 10 microseconds.
In the development of a specific transport navigation system, the use of this or that type of pulsar depends on the specific task, as well as on the visibility of the pulsar at a certain point in time.
Lines 53-67: (in red)
In the USA, Russia and China, the pulsar navigation is mainly used to control space objects. In the event of a failure in the GPS GNSS, the X-ray pulsar navigation system becomes indispensable because it uses the sextant principle as suggested by the project of the US National Space Agency named Neutron-star Interior Composition Explor-er/Station Explorer for X-ray Timing and Navigation Technology (NICER/SEXTANT). This project demonstrates an XNAV pulsar navigation system that is currently used on the spacecraft Artemis [38].
It is known that X-ray and gamma-ray pulsars can only be observed by space crafts, because signals from these pulsars are impossible to be detected on the Earth's surface due to their strong absorption by the Earth's atmosphere. The blocking of GPS GNSS, the all transport ground and air means of transport (ships, planes) will remain without cosmic navigation and ground support. The military XNAV pulsar navigation system will not be available for civilian navigation. For these two reasons, the development of a terrestrial radio pulsar navigation system for transportation is a very current scientific and applied task that awaits its solution.
Lines 130 – 136 (in red)
We only consider the questions of pulsar signal processing, in the proposed functional structure of signal processing in a pulsar transport navigation system. Like GNSS systems, it uses signals from 4 suitable pulsars (out of 50) to position the object. More powerful seconds pulsars can be used, which can be observed with smaller antennas. The TOA measurement error for these pulsars is of the order of tens-hundreds of microseconds. The milliseconds pulsars can be used with the measurement error of TOA is less than 10 microseconds
Lines 192 – 1206 (in red)
Most X-ray pulsars emit very weak signals. The Crab pulsar (PSR B0531+21) is the most powerful pulsar in the X-ray range with the radiation flux density ~ 9.9 • 10-9 erg cm -2s-1 with photon energy 2-10 keV. Other pulsars in this range are much weaker emitters. It is known that X-ray and gamma-ray pulsars can only be observed by spacecrafts, because signals from these pulsars are impossible to be detected on the Earth's surface due to their strong absorption by the Earth's atmosphere.
On Earth's surface, only radio pulsars can be used for navigation. In this case, more powerful seconds pulsars can be used, which can be observed with smaller antennas. However, these pulsars have a pulse emission period of the order of 1 sec (seconds pulsars). The TOA measurement error for these pulsars is of the order of tens-hundreds of microseconds. The other group of pulsars are so-called milliseconds pulsars, the pulse repetition period of which is on the order of several milliseconds. However, these pulsars have less radiation power and therefore require antennas with a larger area for observation. However, the measurement error of TOA is less than 10 microseconds.
Lines 283 – 298 (in red)
Commercial GPS jammers are now readily available on the Internet. A chain reaction of space debris, known as an ablation cascade, or a strong Earth-centered solar storm can interfere with GPS capability. Another serious drawback is the lack of accuracy in the subpolar regions of the Earth, caused by the low inclination of the orbits of GPS satellites (about 55 degrees). Without the use of GPS, the military ships at sea must find alternative ways to get an accurate fix on their position, and the military has just resumed training officers in the lost art of celestial navigation [39].
The U.S. Navy and Department of Defense take cyber threats to technology infrastructure seriously [39] and that is why they ordered companies Strategic & Spectrum Missions Advanced Resilient Trusted Systems (S2MARTS) to develop a navigation system that will push GPS to second plan. In the event of a failure in the GPS GNSS, the X-ray pulsar navigation system becomes indispensable because it uses the sextant principle as suggested by the project of the US National Space Agency named Neutron-star Interior Composition Explor-er/Station Explorer for X-ray Timing and Navigation Technology (NICER/SEXTANT). This project demonstrates an XNAV pulsar navigation system that is currently used on the spacecraft Artemis [38].
Lines 321 – 334 (in red)
It is known that X-ray and gamma-ray pulsars can only be observed by space crafts, because signals from these pulsars are impossible to be detected on the Earth's surface due to their strong absorption by the Earth's atmosphere.
In the event of a magnetic storm, radio jamming or other blocking of GPS GNSS, other GNSS radio navigation systems, as well as blocking of ground navigation systems, the all transport ground and air means of transport (ships, planes) will remain without cosmic navigation and ground support. In these situations, the military XNAV pulsar navigation system will not be available for civilian navigation. But even in the absence of any threats to space and ground navigation and communication systems of the 4th and 5th generation, the presence of a pulsar navigation system will be an additional source of navigational information in certain areas of the earth's surface.
For these two reasons, the development of a terrestrial radio pulsar navigation system for transportation is a very current scientific and applied task that awaits its solution.
Lines 600 – 639 (in red)
The article is devoted to the theoretical aspects regarding the development of an autonomous transport radio navigation system with a small receiving antenna, using radio signals from pulsars, similar to navigation systems for space navigation. Therefore, the proposed structure of a transport navigation system is similar to that of a spacecraft. It should consist input (X-ray and radio pulsar signals), three main blocks - instruments, timing estimation and location algorithm [2], and output (position estimates). In this paper, only the issues of processing pulsar signals are considered, and that is related to the input and the first two blocks of the navigation system. This is also evident from the proposed functional structure of signal processing in a pulsar transport navigation system presented on Fig.13.
In accordance with [2], we propose that in a transport navigation system, the antenna system should synthesize beams directed at different directions, where the used radio pulsars are located. Zeros are formed in the diagram of reception of the used pulsar signals in order to remove noise sources such as the sun. That requires an antenna geometry based on single antennas, each with its own receiver, and processing the received signal from the observed pulsar [2].
On Earth's surface, only radio pulsars can be used for transport navigation. Like GNSS systems, it uses signals from 4 suitable pulsars (out of 50) to position the object. These pulsars are not uniformly distributed, but are grouped in certain directions (at least 6 clusters can be determined) [2].
For transport navigation, more powerful seconds pulsars can be used, which can be observed with smaller antennas. These pulsars have a pulse emission period of the order of 1 sec (seconds pulsars). The TOA measurement error for these pulsars is of the order of tens-hundreds of microseconds. The milliseconds pulsars, the pulse repetition period of which is on the order of several milliseconds. These pulsars have less radiation power and therefore require antennas with a larger area for observation, and the measurement error of TOA is less than 10 microseconds [41,42].
Geometric reduction of accuracy in pulsar navigation requires that the positioning reference sources are widely distributed in angles of arrival. If the observed group of pulsars is large, better accuracy characteristics will be achieved (in a single-channel and single-antenna observation). But at the same time, the minimum integration time for each individual window will increase, along with the minimum time for getting an estimate of the first position. Therefore, when choosing the pulsars for observation, there should be trade-offs whether to select a group of pulsars achieving higher positioning accuracy with sufficient time for integration or a group of pulsars requiring less time for observation with lower accuracy.
The strategy of the transport navigation algorithm is first to group several pulsars (four) in order to use small accumulation time and obtain an average position accuracy, and then to include more pulsars for improving the position accuracy.
Lines 915 – 921 (in red)
In this case, more powerful seconds pulsars can be used, which can be observed with smaller antennas. However, these pulsars have a pulse emission period of the order of 1 sec (seconds pulsars). The TOA measurement error for these pulsars is of the order of tens-hundreds of microseconds. The other group of pulsars are so-called milliseconds pulsars, the pulse repetition period of which is on the order of several milliseconds. However, these pulsars have less radiation power and therefore require antennas with a larger area for observation. However, the measurement error of TOA is less than 10 microseconds.
